# An Improved Active Damping Method for Enhancing Robustness of LCL-Type, Grid-Tied Inverters under Weak Grid Conditions

**DOI:** 10.3390/s23198203

**Published:** 2023-09-30

**Authors:** Shanwen Ke, Yuren Li

**Affiliations:** School of Automation, Northwestern Polytechnical University, Xi’an 710129, China; kswkch@mail.nwpu.edu.cn

**Keywords:** LCL filter, ideal differentiator, digital low-pass filter, digital control, active damping

## Abstract

The conventional proportional-gain-feedback link can only obtain the smallest effective damping region (EDR) due to the control delay among all the active damping methods regarding the capacitor current feedback. The digitally controlled system tends to be unstable when the system resonant frequency reaches the critical frequency caused by the grid impedance variation. To weaken the adverse effect on the system caused by the control delay, phase-lead feedback links are applied along the feedback path to provide phase compensation. By taking the simplicity and reliability of the feedback links into account, this paper proposes an alternative to an ideal differentiator, which consists of the Tustin discrete form of ‘*s*’ and a digital low-pass filter. This proposed method has an identical phase frequency characteristic as an ideal differentiator but a better magnitude frequency characteristic, and its EDR can reach [0, *f*_s_/3]. The system stability analysis is conducted under different resonant frequencies, and under the condition of a weak grid, the co-design approach of the active damper and digital controller is presented. Finally, the experimental results are shown to verify the proposed method.

## 1. Introduction

At present, the main way to develop renewable energy on a large scale is distributed generation based on renewable energy. As the energy conversion interface between the renewable energy power generation unit and the power grid, a grid-connected inverter is used to convert DC energy into high-quality AC and feed it into the power grid. Generally, a suitable filter needs to be added between the inverter and the power grid. Compared with the L filter, the LCL filter contains filter capacitance, so it has smaller volume and stronger high-frequency filtering ability [1,2,3,4]. However, LCL filters introduce a pair of conjugate poles on the closed-loop stability boundary, which is likely to impair system stability [5,6]. To address the issue, various active damping methods are usually used [7,8].

Active damping techniques are mainly divided into two kinds including the single-loop structure of the grid current feedback and the dual-loop structure of the grid current plus capacitor current feedback. The former means fewer sensors and software-based observers are needed, for which the single-loop current control schemes have increasingly been studied [9,10,11,12,13,14,15,16]. Reference [15] shows that a stable single-loop control scheme of the grid current without damping is implemented for the reason that an inherent damping characteristic is introduced when the digital delay is considered. The inherent damping characteristic is, however, available in the frequency region beyond one sixth of the system sampling frequency. The system will become unstable as a result of the wide variation of the resonance frequencies in the weak grid. Reference [16] proposed an active damping technique with a negative high-pass filter along the grid current feedback path, which helps to mitigate the phase lag caused by time delays found in the digital implement. A first-order, high-pass filter can only provide phase compensation less than π/2. In order to attain bigger phase compensation, hence bigger EDR, [17] introduced an improved active damping scheme by adding a delay link along the damping loop mentioned in [16], which is equivalent to cascading one more first-order, high-pass filter. A second-order, high-pass filter can provide sufficient phase compensation to counteract the negative influence caused by time delay, and hence, enlarge the EDR. Although only one quantity needs to be measured and fewer sensors are needed in single-loop techniques, the system performance can be affected due to variation of the parameters. The latter is commonly used in practice for its excellent damping effect and simple implementation [18,19,20,21,22,23]. The traditional capacitor current feedback active damping (CCFAD) adopts a proportional gain feedback of the capacitor current, which establishes an EDR in the frequency range [0, *f*_s_/6] due to the control delay. Here, *f*_s_/6 is defined as the critical frequency between the positive and negative damping region [24]. As a result, a lot of literature is devoted to increasing EDR through different strategies that improve the capacitor current feedback loop. A second-order-generalized-integrator-based time-delay compensation method for extending the stable region is presented in [25], which can compensate a maximum delay of a half-sampling period and the EDR can be increased up to *f*_s_/4. Reference [26] introduces a first-order leading link and [27] proposes a delayed feedback link into the capacitor current feedback loop, respectively. Though both methods achieve some phase compensation, they have a relatively small EDR, i.e., [0, *f*_s_/4]. Many efforts have been focused on achieving better frequency derivative characteristics whose corresponding critical frequency is *f*_s_/3, aiming to further increase the critical frequency. A virtual *RC* damping method is proposed in [28], where the extra virtual capacitance and inductance benefit the delay compensation and the EDR is enlarged up to [0, *f*_s_/3]. References [29,30] introduce the derivative term to counteract the time delay; however, the digital implement of the proposed method is not discussed. A first-order, high-pass filter is applied in [31] to mimic derivative features, which have a better disturbance rejection capability. The phase compensation amount can be modified by varying the high-pass filter parameters, hence changing the size of the critical frequency. However, a large phase lag error means a smaller EDR whose critical frequency is above *f*_s_/4, but away from *f*_s_/3. The nonideal generalized integrator (nGI) [32] has so far proven to be the best way to realize the highly accurate derivative. Reference [33] proposes two digital differentiators, as simple alternatives of nGI, which are first-order differentiators based on a backward Euler plus digital lead compensator and a second-order differentiator based on a Tustin plus digital notch filter. These methods can obtain better differential characteristics at higher frequencies; hence, the critical frequencies are closer to *f*_s_/3 in comparison with the high-pass filter. However, their critical frequencies cannot approach *f*_s_/3 due to the small phase lag in the mid- and high-frequency domains. Accordingly, the damping limitation and stability challenge will occur at the critical frequency *f*_s_/3 [34]. Reference [35] introduces a new feedback link of capacitor–current based on the recursive infinite impulse response digital filter, which is equivalent to a high-order, high-pass filter with sufficient phase lead compensation. The EDR is extended close to the maximum value *f*_s_/2. However, the high-order filter in the digital control is unreliable and imposes complexity on the control system. The phase compensation link used in [36] is a second-order, high-pass filter whose magnitude in the middle and low frequency domains is almost zero, and it can provide a greater phase lead. However, it ignores the frequency characteristic of the second-order filter in the high-frequency region, especially near the Nyquist frequency.

This article proposed an alternative of the ideal differentiator, which consists of the ideal differentiator’s Tustin discrete form and a digital low-pass filter. The proposed methods are consistent with the ideal differentiator in phase characteristics, which ensure the phase compensation can achieve π/2; namely, the corresponding EDR is [0, *f*_s_/3]. Also, the magnitude characteristics of the proposed method stay approximately constant in the middle- and low-frequency region and are attenuated greatly in the high-frequency region, which ensures the noise rejection capability.

This paper is organized as follows: Section 2 introduces the system model and conducts the impedance-based analysis in the continuous *s*-domain; Section 3 depicts the property of the proposed method in the discrete *z*-domain and deduces the discretization or digitally controlled system; Section 4 conducts the stability analysis of the system with the proposed active damping method; Section 5 introduces the co-design procedure of the digital controller and active damper; and Section 6 presents the experimental results and validates the theoretical analysis. Finally, the conclusion of this paper is shown in Section 7.

## 2. Continuous S-Domain Analysis

### 2.1. System Description

Figure 1 shows the general structure of a three-phase, grid-connected, voltage-source inverter (VSI) with an LCL filter and constant *DC*-link voltage *V*_dc_. For the sake of analysis simplicity, the following two assumptions are considered valid, one of which is that the parasitic resistances of the circuit are neglected, which results in the worst damping condition, and the other of which is that the grid voltage is assumed to be balanced. In Figure 1, *L*_g_ represents the grid inductance which is variable, *G*_h_(*s*) is the feedback block of capacitor current *i*_c_, and *G*_PR_(*s*) is the current regulator that generally applies a proportional plus resonance (PR) controller, which can provide substantial gain at fundamental frequency; the transfer function is given by (1) [37].
(1)$$G_{\mathrm{PR}}(s)=K_{P}+\frac{2K_{r}\omega_{i}s}{s^{2}+2\omega_{i}s+\omega_{o}^{2}}$$
where *K*_P_ and *K*_r_ are proportional and resonant coefficients, respectively, and *ω*_i_ is the −3 dB bandwidth for which generally *ω*_i_ =π rad/s [38]. For such current regulators, the steady-state tracking error is substantially eliminated at the fundamental frequency *ω*_o_.

Figure 2 depicts the per-phase block diagrams of the LCL-type, grid-tied inverter system with CCFAD. *i*_2_ is the measured grid current; $i_{2}^{*}$ is the command grid current which is set according to the system request; *i*_c_ is the measured capacitor current, and the inverter is modeled as a linear gain *K*_PWM_ with one and half sampling period digital delay *G*_d_(*s*) which is composed of a computation delay and a modulation delay. The expression of *G*_d_(*s*) is given as follows [39]:(2)$$G_{d}(s)=e^{-1.5sT_{s}}$$
where *T*_s_ is the system sampling period.

### 2.2. Impedance-Based Analysis

To better demonstrate the damping mechanism of the active damping method realized by the capacitor current feedback and keep the system closed-loop characteristics unchanged, some modifications are introduced on the model shown in Figure 2a and the equivalent block diagram is shown in Figure 2b. Obviously, it can be seen that the feedback coefficient of the capacitor current is equivalent to be an impedance *Z*_eq_ paralleled with the capacitor. As shown in Figure 3, a virtual inductance *X*_eq_ and a virtual resistance *R*_eq_ are connected in parallel and constitute impedance *Z*_eq_, which is represented by (3)
(3)$$Z_{\mathrm{eq}}=\frac{L_{1}}{CK_{\mathrm{PWM}}G_{h}(s)}e^{1.5sT_{s}}=R_{\mathrm{eq}}(\omega )//jX_{\mathrm{eq}}(\omega )$$
where *G*_h_(*s*) is the feedback coefficient, which has all-pass or high-pass filter (HPF) characteristics due to *i*_c_ being the high-frequency current signal. For analysis simplicity, the first-order HPF are studied in this paper. The general expression of first-order HPF can be given as follows:(4)$$G_{h}(s)=\frac{K_{d}s}{s+\omega_{d}}=Ae^{j\theta },\theta \in (0,\pi /2)$$
where *K*_d_ represents the gain of HPF, *ω*_d_ represents the cutoff frequency of the HPF. Transformed into the polar coordinate form, *G*_h_(*s*) consists of two parts presented in (4), where *A* is the modulus and *θ* is the lead-phase angle with the boundary values 0 and π/2 not included in the range available. Substituting (4) into (3) and doing some simple mathematical operation, (5) can be attained.
(5)$$Z_{\mathrm{eq}}=R_{A}e^{\left(1.5sT_{s}-\theta \right)}=R_{\mathrm{eq}}\left(\omega \right)//jX_{\mathrm{eq}}(\omega )$$

Substituting *s = jω* into (5), the expression of *R*_eq_, *X*_eq_ can be attained in the frequency domain.
(6)$$\begin{array}{l} R_{\mathrm{eq}}\left(\omega \right)=R_{A}/\cos(1.5\omega T_{s}-\theta )\\ X_{\mathrm{eq}}(\omega )=R_{A}/\sin(1.5\omega T_{s}-\theta ) \end{array}$$
where *R*_A_ = *L*_1_/(*CK*_PWM_*A*), which is the equivalent resistance of CCFAD in analog control.

As shown in (6), the real term *R*_eq_(*ω*) and the imaginary term *X*_eq_(*ω*) can both become negative after introducing the finite delay time and lead phase angle. The equivalent resonant frequency $\omega_{r}^{\prime }$ will make a difference compared with the actual resonant frequency *ω*_r_ after introducing the imaginary term *X*_eq_(ω).
(7)$$\omega_{r}^{\prime }=\sqrt{\frac{L_{1}+L_{2}+L_{g}}{L_{1}(L_{2}+L_{g})C}}.\sqrt{\frac{X_{\mathrm{eq}}}{X_{\mathrm{eq}}-1/\omega C}}=\omega_{r}^{}.\sqrt{\frac{X_{\mathrm{eq}}}{X_{\mathrm{eq}}-1/\omega C}}$$

*X*_eq_(*ω*) acts like an inductance when *X*_eq_(*ω*) is positive, which counteracts the effect of the capacitor; the resonant frequency $\omega_{r}^{\prime }$ thus becomes a little higher than *ω*_r_. On the contrary, *X*_eq_(*ω*) acts like a capacitor when *X*_eq_(*ω*) is negative, which enhances the effect of the capacitor; the resonant frequency $\omega_{r}^{\prime }$ thus becomes a little lower than *ω*_r_.

The negative real term causes the nonminimum-phase behavior, which adds open-loop right half-plane poles (RHP) to the current control loop and thus impairs the system robust stability. From (6), to be positive for *R*_eq_(*ω*), the critical frequency *f*_crit_ can be attained as follows:(8)$$f_{\mathrm{crit}}=f_{s}/6+\theta f_{s}/\left(3\pi )\;\;\theta \in (0,\;\pi /2\right)$$

As shown in (8), when *θ* changes from 0 to π/2, *f*_crit_ increases from *f*_s_/6 to *f*_s_/3. Figure 4 plots the frequency region of *R*_eq_(*ω*) being positive as *θ* changes. Especially, taking the two boundary values into account, it is seen that when *θ* = 0, *f*_crit_ = *f*_s_/6, and when *θ* = π/2, *f*_crit_ = *f*_s_/3. Therefore, the EDR obtained is [0, *f*_s_/3) according to (8).

## 3. Discrete Z-Domain Analysis

### 3.1. Improved Active Damping Method

In order to study the damping mechanism of the feedback link in a digitally controlled system, the discrete form of *G*_h_(*s*) is needed. The Tustin discrete expression of the feedback link *G*_h_(*s*) is shown in (9).
(9)$$G_{h}(z)=\frac{2K_{d}(z-1)}{\left(2+\omega_{d}T_{s})z+(\omega_{d}T_{s}-2\right)}\;=H\cdot \frac{z-1}{z-a}$$
where *H* = 2 *K*_d_/(2 + *ω*_d_*T*_s_), *a* = (*ω*_d_*T*_s_ − 2)/(*ω*_d_*T*_s_ + 2), (−1 < *a* < 1).

Figure 5 depicts the zero-pole distribution of *G*_h_(*z*), in which there is a zero *z* = 1 and a pole *z* = *a*, and the pole *a* slides on the real axis within the range (−1, 1), which corresponds with each different lead phase angle *θ* within the range (0, π/2). As of *G*_h_(*s*), the two boundary values 0 and π/2 are not available. However, unlike the *G*_h_(*s*), the parameter *a* of *G*_h_(*z*) can change within an extended range [−1, 1]. *G*_h_(*z*) simplifies to *H* and *f*_crit_ = *f*_s_/6 in the case of *a* = 1, which is just the conventional CCFAD, and it can be attained that *G*_h_(*z*) = *H* (*z* − 1)/(*z* + 1) in the case of *a* = −1, which is similar to the Tustin discrete form of an ideal differentiator. As seen in Figure 6a, the ideal differentiator can provide π/2 phase lead compensation and finite gain over the entire frequency domain, but can not be implemented in practice. On the contrary, *G*_h_(*z*) (*a* = −1) can achieve the same phase-lead compensation and be easily realized in the digital-control system, but infinite gain arises at the Nyquist frequency, which is more likely to amplify the figure high-frequency noises. In order to overcome the drawback of *G*_h_(*z*) (*a* = −1), this paper proposes a strategy of cascading a digital low-pass filter with it.

A digital low-pass filter is cascaded with *G*_h_(*z*) (*a* = −1), aiming to suppress the high-frequency infinite gain of *G*_h_(*z*) (*a* = −1) and keep the other frequency characteristics approximately unchanged. The discrete expression of the digital low-pass filter is given as [26]:(10)$$G_{c}(z)=\frac{z+m+z^{-1}}{2+m}$$
where *m* is an adjustable parameter. Substituting *z* = *e^sT^* into (10), the continuous expression of *G*_c_(*s*) is obtained as (11)
(11)$$G_{c}(s)=\frac{e^{sT}+m+e^{-sT}}{2+m}=\frac{2\cos(\omega T)+m}{2+m}$$

*G*_c_(*s*) is positive when *m* ≥ 2 because cos(*ωT*_s_) ϵ [−1, 1] and *G*_c_(*s*) behaves as a variable constant over the whole frequency domain, which means its phase is always zero. In order to ensure the gain stays zero at the Nyquist frequency, let *m* = 2 and cos(*ω*_s_*T*/2) = −1; hence, *T* = *T*_s_. The improved discrete form of *G*_h_(*z*) (*a* = −1) after cascading *G*_c_(*s*) is obtained as
(12)$$G_{h}^{\prime }(z)=G_{h}(z)G_{c}(z)=\frac{z-1}{z+1}\cdot \frac{z+2+z^{-1}}{4}$$

For comparison purposes, multiply the several functions in Figure 6 by a different constant to ensure that the magnitude–frequency characteristics are almost the same in the low-frequency band without changing the phase–frequency characteristics. As seen in Figure 6a, obviously, *G*_c_(*s*) actually is a digital low-pass filter, whose phase keeps zero in the entire frequency band, and it has little effect on the magnitude–frequency characteristic of *G*_h_(*z*) (*a* = −1) in the low frequency band after cascading it. However, the gain of *G*_h_(*z*) (*a* = −1) in the high frequency band, especially at the Nyquist frequency, is attenuated substantially to below 0 dB, referring to the frequency characteristic curve of $G_{h}^{\prime }(z)$. Reference [33] proposes a first-order differentiator based on a backward Euler plus lead compensator and a second-order differentiator based on a Tustin plus digital notch filter to be alternatives of an ideal differentiator, whose transfer functions are marked as *G*_back-lead_(*z*) and *G*_tustin-DNF_(*z*), respectively. In addition to $G_{h}^{\prime }(z)$ and the ideal differentiator, the bode plots of *G*_back-lead_(*z*) and *G*_tustin-DNF_(*z*) are also shown in Figure 6b. It is clear that the magnitudes of *G*_back-lead_(*z*) and *G*_tustin-DNF_(*z*) are finite and relatively higher compared with the ideal differentiator in the high-frequency domain, and the magnitude of $G_{h}^{\prime }(z)$ is almost flattened in the entire frequency range except for approximate 0 dB at the Nyquist frequency, implying better disturbance rejection ability. The phases of *G*_back-lead_(*z*) and *G*_tustin-DNF_(*z*) are almost the same, with a slight lag in the mid- and high-frequency domain compared with the ideal differentiator. However, the phase frequency characteristic of $G_{h}^{\prime }(z)$ is totally identical to that of the ideal differentiator. Based on the analysis above, it is concluded that $G_{h}^{\prime }(z)$ is the closest match with the ideal differentiator in terms of frequency characteristics. In addition, the EDR obtained is [0, *f*_crit_] (*f*_crit_ < *f*_s_/3) using the proposed method in [33] due to the phase lag in the mid- and high- frequency domain, while the EDR obtained is [0, *f*_s_/3] using the proposed method in this paper. The feature comparisons of different digital compensation methods are listed in Table 1.

For the theoretical analysis simplicity, only using *G*_h_(*z*) without considering the correction link *G*_c_(*z*) is sufficient in analyzing the system stability and introducing the design procedure of the digital controller and damper in the following.

### 3.2. System Discretization

System analysis based on impedance equivalence is done in the *s*-domain, which is appropriate with the discrete controller not included. To co-design the discrete current controller and active damper, it is more accurately done in the *z*-domain. The continuous system model in Figure 2 is transformed into a discrete one, shown in Figure 7.

Digital pulse width modulation (DPWM) is modeled as a zero-order holder (ZOH) with half a period delay, which is proven to be an appropriate approximation of the uniformly sampled DPWM [40]. The next block is the *z*^−1^ delay block which represents a period computation delay. The transfer function *Y*_g_(*s*) representing the LCL filter is given in (13) and its discrete counterpart showed in (14), respectively.
(13)$$Y_{g}(s)=\frac{i_{2}}{V_{\mathrm{inv}}}\left|{}_{V_{g}=0}\right.=\frac{1}{sL_{1}(L_{2}+L_{g})C}\frac{1}{s^{2}+\omega_{r}^{2}}$$
(14)$$\begin{array}{cc} \frac{i_{2}(z)}{v_{M}^{\prime }(z)} & =Z\left[G_{\mathrm{ZOH}}(s)K_{\mathrm{PWM}}\cdot Y_{g}(s)\right]\\ & =\;Z\left[\frac{1-e^{-sT_{s}}}{s}\frac{K_{\mathrm{PWM}}}{sL_{1}(L_{2}+L_{g})C}\frac{1}{s^{2}+\omega_{r}^{2}}\right]\\ & =\frac{K_{\mathrm{PWM}}T_{s}}{\left(L_{1}+L_{2}+L_{g})(z-1\right)}\\ & \;-\frac{K_{\mathrm{PWM}}(z-1)\sin(\omega_{r}T_{s})}{\omega_{r}(L_{1}+L_{2}+L_{g})[z^{2}-2z\cos(\omega_{r}T_{s})+1]} \end{array}$$
where *G*_ZOH_ = (1 − *e^−sT^*^s^)/*s* denotes the transfer function of the ZOH. After applying Tustin transformation, the discrete expression of the PR current controller *G*_PR_(*s*) in (1) can be obtained shown in (15).
(15)$$G_{\mathrm{PR}}(z)=K_{P}+\frac{K_{r}\sin(\omega_{i}T_{s})}{2\omega_{i}}\frac{z^{2}-1}{z^{2}-2z\cos(\omega_{i}T_{s})+1}$$

According to (9), (14) and (15), the expression of the transfer function for the dual-loop control scheme can be derived as follows:(16)$$\begin{array}{l} T_{D}(z)=\frac{H_{i2}G_{\mathrm{PR}}(z)K_{\mathrm{PWM}}}{\omega_{r}(L_{1}+L_{2}+L_{g})(z-1)}\cdot \\ \;\;\;\frac{\omega_{r}T_{s}[z^{2}-2z\cos(\omega_{r}T_{s})+1]-{\left(z-1\right)}^{2}\sin(\omega_{r}T_{s})}{z[z^{2}-2z\cos(\omega_{r}T_{s})+1]+\frac{{\left(z-1\right)}^{2}}{\left(z-a\right)}\frac{HK_{\mathrm{PWM}}\sin(\omega_{r}T_{s})}{\omega_{r}L_{1}}} \end{array}$$

From (16), it can be concluded that there is no nonlinear link like transcendental functions in *z*-domain compared with that in *s*-domain. The eigenvalues can thus be solved simply, hence obtaining the condition of including RHP poles in *T*_D_(*z*).

## 4. System Stability Analysis

System stability needs to be further analyzed for the cases of different resonant frequencies after introducing the feedback link *G*_h_(*z*) = *H*(*z* − 1)/(*z* + 1). Reference (16) can be redrawn as (17).
(17)$$\begin{array}{l} T_{D}(z)=\frac{H_{i2}G_{\mathrm{PR}}(z)K_{\mathrm{PWM}}}{\omega_{r}(L_{1}+L_{2}+L_{g})(z-1)}\cdot \\ \;\;\;\frac{\omega_{r}T_{s}[z^{2}-2z\cos(\omega_{r}T_{s})+1]-{\left(z-1\right)}^{2}\sin(\omega_{r}T_{s})}{z[z^{2}-2z\cos(\omega_{r}T_{s})+1]+\frac{{\left(z-1\right)}^{2}}{\left(z+1\right)}\frac{HK_{\mathrm{PWM}}\sin(\omega_{r}T_{s})}{\omega_{r}L_{1}}} \end{array}$$
where *G*_PR_(*z*) has two conjugate poles on the unit circle that does not belong to the right-half plane. To judge whether the system has the RHP poles, the poles distribution of *T*_D_(*z*), more accurately, the roots distribution of denominator *D*(*z*) of *T*_D_(*z*) whose expression is given in (18) should be analyzed.
(18)$$D(z)=z[z^{2}-2z\cos(\omega_{r}T_{s})+1]+\frac{{\left(z-1\right)}^{2}}{{\left(z+1\right)}^{2}}\frac{HK_{\mathrm{PWM}}\sin(\omega_{r}T_{s})}{\omega_{r}L_{1}}=0$$

According to the Routh criterion, the number of RHP poles of (16) is equivalent to the changing times of the first-column elements [*a*_0_ *a*_1_ *b*_1_ *c*_1_ *a*_4_] of (A2) in Appendix A being positive or negative. Considering *f*_r_ < *f*_s_/2, then *ω*_r_*T*_s_ < π and *a*_0_ > 0, *a*_4_ > 0, and assuming that *a*_1_ *b*_1_ *c*_1_ are all above zero, parameter *H* must satisfy the following three requirements shown in (22).

The parameter *H* must satisfy the following three requirements shown in (19) by conducting the Routh criterion on *D*(*z*). When *ω*_r_*T*_s_ ranges (0,π), the second parts of the functions *H*_1_, *H*_2_, *H*_3_ in (19) are plotted in Figure 8, in which it can be concluded that *H*_3_ < *H*_2_ < *H*_1_.
(19)$$\left\{\begin{array}{l} H<\frac{\omega_{r}L_{1}}{K_{P}\sin(\omega_{r}T_{s})}\cdot \frac{1+\cos(\omega_{r}T_{s})}{2}=H_{1}\\ H<\frac{\omega_{r}L_{1}}{K_{P}\sin(\omega_{r}T_{s})}\cdot \frac{\sqrt{1+2{(1+\cos(\omega_{r}T_{s}))}^{2}}-1}{2}=H_{2}\\ H<\frac{\omega_{r}L_{1}}{K_{P}\sin(\omega_{r}T_{s})}\cdot \frac{1+2\cos(\omega_{r}T_{s})}{3}=H_{3} \end{array}\right.$$

The relationship between the value range of *H* and the number of RHP poles of the system is listed in detail in Table 2.

For analysis simplicity, only the case *H* > 0 is studied in this paper. From Table 2, it is observed that the system has no RHP poles when *H* < *H*_3_, whereas there are two RHP poles when *H* > *H*_3_. Therefore, *H*_3_ is the critical value and the critical frequency *f*_crit_ = *f*_s_/3 can be solved when *H*_3_ = 0, which just matches the critical frequency *f*_crit_ (*a* = −1) in the analysis of virtual resistance in the previous section. It is concluded that the condition of the system applying the feedback link *G*_h_(*z*) = *H*(*z* − 1)/(*z* + 1) with no RHP poles is *f*_r_*’* < *f*_crit_; in other words, *R*_eq_(*ω*) is positive at the frequency below *f*_crit_.

For convenience, the method proposed in this paper is called Method 1 and the CCFAD is called Method 2. The compared analysis of the two methods is presented in the following.

The system in *z*-domain is commonly evaluated with the Nyquist stability criterion, i.e., *P* = 2(N+ − N−), where *P* denotes the number of the open-loop RHP poles, and N+ and N- denote the numbers of positive and negative −180° crossings, respectively. It would be counted as one positive or negative crossing as long as the gain is above 0 dB when the phase transits the −180°.

Based on the previous analysis, the critical frequencies *f*_crit_ are *f*_s_/6 and *f*_s_/3 for method 1 and method 2, respectively; as a result, the stability requirement of the two methods are different. As listed in Table 3, the cases are classified in terms of the frequency band on which the equivalent resonant frequency *f*_r_’ is located. As for the method 1, when *f*_r_’ < *f*_s_/6, *R*_eq_(*f*_r_’) is positive, i.e., there are no RHP poles and just one negative −180° crossing at *f*_r_; when *f*_r_’ > *f*_s_/6, *R*_eq_(*f*_r_’) is negative, i.e., there are two RHP poles and −180° crossing at *f*_r_ and *f*_s_/6, respectively. Here, GM_1_ and GM_2_ represent the gain margin at *f*_r_ and *f*_s_/6, respectively. Especially, when *f*_r_’= *f*_s_/6, *R*_eq_(*f*_r_’) is negative and there is no −180° crossing, which leads to system instability due to the fact that the requirements of GM_1_ and GM_2_ cannot be met simultaneously. As for method 2, when *f*_r_’ < *f*_s_/3, *R*_eq_(*f*_r_’) is positive, i.e., there are no RHP poles and just one negative −180°crossing at *f*_1_; when *f*_r_’ ≥ *f*_s_/3, *R*_eq_(*f*_r_’) is negative, i.e., there are two RHP poles and −180°crossing at *f*_1_ and *f*_2_, respectively. Here, GM_1_ and GM_2_ represent the gain margin at *f*_1_ and *f*_2_, respectively.

As listed in Table 3, for either method 1 or method 2, when *f*_r_’ < *f*_crit_, the gain margin is required to be above 0 dB. On the contrary, the gain margin requirement is more stringent when *f*_r_’ > *f*_crit_; therefore, a larger EDR, and in other words, a larger *f*_crit_, is desirable.

In a weak grid, the grid impedance *L_g_* is variable, which leads to the variation of *f*_r_, hence weakening the system robust stability. The compared analysis of the two methods under *L_g_* variation is presented in the following.

Parameters used for simulation and experiment are given in Table 4, where two different capacitor values are chosen to generate different filter resonance frequencies on the condition that convert-side inductance *L*_1_ and grid-side inductance *L*_2_ are selected unchanged. For the weak grid, the grid inductance *L_g_* may change from zero to the maximum value available, which is chosen corresponding to the short circuit ratio (SCR) of 10 [5]. Compared with the inductance, the commercial capacitors are more accurate and well-tested; therefore, changing the capacitor’s value for analysis is a preferable choice to inductance value.

Figure 9 illustrates the frequency response of the two methods mentioned above, which is affected by the variation of the grid inductance *L_g_*. For the case *C* = 9.4 μF and *L_g_* = 0 mH, the original resonant frequency *f*_r_ is equal to 2.01 kHz, which is a little above and close to *f_s_*/6. As shown in Figure 9a, when method 1, namely *G*_h_(*z*) = *H,* is applied, referring to the curve 1, the forward-path phase transits through −180° at *f_s_*/6 and *f*_r_, respectively. The requirement of the gain margin GM_1_ < 0 at *f_s_*/6 and GM_2_ > 0 at *f*_r_ must be met to ensure the system stability according to the Nyquist stability criterion. When *L_g_* varies from 0 mH to 0.8 mH dynamically, the actual *f*_r_ will get closer to and finally be approximately equal to *f_s_*/6. Obviously, it can be seen from curve 2 in the plot that the two frequency points *f_s_*/6 and *f*_r_ are so close (almost overlap) that the requirement of the gain margin GM_1_ and GM_2_ cannot be met simultaneously or the value of GM_1_ and GM_2_ are not adequate to ensure system robust stability. When *L_g_* increases further up to 2.0 mH, *f*_r_ thus shifts below and away from *f*_s_/6. The forward-path phase transits through −180° only at *f*_r_, and the system goes back to the stability state again in case the requirements for PM and GM_1_ are met simultaneously, as illustrated by curve 3 in the plot. As shown in Figure 9b, when method 2, namely *G*_h_(*z*) = *H*(*z* − 1)/(*z* + 1) is applied and *L_g_* varies within the range (0, 2 mH), the forward-path phase transits through −180° once at the frequency below *f*_s_/3, for which the system stability can be guaranteed only when the requirements for PM and GM_1_ are met simultaneously.

For the case *C* = 3.9 *μ*F and *L_g_* = 0 mH, the resonant frequency *f*_r_ is equal to 3.12 kHz, which is a little below and close to *f_s_*/3. Applying method 2, referring to Table 2, when the coefficient *H* < *H*_3_ mentioned in the previous section, *R*_eq_(*f*_r_’) is positive and the forward-path phase transits through −180° only at the frequency *f*_1_; when the coefficient *H* > *H*_3_, *R*_eq_(*f*_r_’) is negative and the forward-path phase transits through −180° at *f*_1_ and *f*_2_, respectively, as illustrated by curve 2 and curve 1 in Figure 9c. However, one crossing point through −180° will be gone and curve 1 turns into curve 3 (*L_g_* = 2.0 mH) eventually when *L_g_* changes from 0 to 2 mH.

To summarize the above analysis, it can come to some conclusions as follows:

For the active damping method with *G*_h_(*z*) = *H*, an issue that the system will be unstable or have low robustness when the two −180°crossing points *f*_r_ and *f*_s_/6 almost overlap is likely to arise due to the variable grid inductance *L_g_*; this is inevitable in the weak grid.

For the active damping method with *G*_h_(*z*) = *H*(*z* − 1)/(*z* + 1) whose effects on the system can be divided into two parts: (*z* − 1)/(*z* + 1) and *H*, the first part decides the critical frequency *f*_s_/3 is much above *f*_s_/6; hence, there just exists one −180° crossing point when *f*_r_*’* is lower than *f*_s_/3, which is easier to satisfy the system stability requirements. Moreover, it is worth noting that *f*_1_ < *f_s_*/6 and *f*_2_ > *f_s_*/3 due to phase compensation at *f_s_*/6 and *f*_s_/3 caused by *G*_h_(*z*). For example, the phase compensation angle is π/2 at *f*_r_ referring to (20) obtained by substituting *z* = *e^jω^*^r*T*s^ into (17). In the meantime, through tuning the second part *H*, the appropriate damping and sufficient stability margin can be achieved.
(20)$$T_{D}(e^{j\omega_{r}T_{s}})=\frac{K_{P}L_{1}\mathrm{ctg}(\omega_{r}T_{s}/2)}{H(L_{1}+L_{2}+L_{g})}e^{-j(\pi +\pi /2)}$$
where *K*_p_ denotes the proportional efficiency of the PR controller, which can simplify to *K*_p_ in the frequency region above the cut-off frequency *f*_c_.

## 5. Co-Design of the Active Damper and Current Controller

The conventional approach to the design and control system with an active damper is to design the current controller first and then design the active damper, which has ignored the virtual impedance effect on current controller. A reasonable design procedure for the overall control strategy is to co-design the current controller and the active damper simultaneously using root locus or other analytical techniques. Therefore, the approach proposed in this paper is that the constraint of the system stability margin requirement on *H* is derived first followed by the constraint of the variable grid impedance *L_g_* on *H*, and the suitable *H* is finally selected after considering all constraints.

According to Figure 2, the expression of the open-loop transfer function of the system model in the *s*-domain can be obtained.
(21)$$\begin{array}{l} T_{D}(s)=\frac{K_{\mathrm{PWM}}}{L_{1}C(L_{2}+L_{g})}\cdot \\ \;\;\;\;\frac{G_{\mathrm{PR}}(s)e^{-1.5sT_{s}}}{s(s^{2}+s\frac{H_{i1}K_{\mathrm{PWM}}e^{-1.5sT_{s}}}{L_{1}}G_{h}(s)+\omega_{r}^{2})} \end{array}$$
where the expression of *G*_h_(s) is given as (22) by substituting the *z* = *e*^s*T*s^ into *G*_h_(s).
(22)$$G_{h}(s)=H\frac{e^{sT_{s}}-1}{e^{sT_{s}}+1}$$

When analyzing the amplitude frequency characteristic of *T*_D_(s), the filter capacitor can be neglected below the cut-off frequency *f*_c_ due to its reactance being far higher than that of the grid-side inductance, and *G*_PR_(s) can be simplified to *K*_P_ at *f*_c_ and *K*_P_ + *K*_r_ at fundamental frequency *f*_o_. Therefore, (23) and (24) can be obtained by substituting *s* = *j*2π*f*_c_ and *s* = *j*2π*f*_o_ into (21), respectively.
(23)$$\left|T_{D}(j2\pi f_{c})\right|\approx \left|\frac{H_{i2}K_{\mathrm{PWM}}K_{P}}{2\pi f_{c}(L_{1}+L_{2}+L_{g})}\right|=1$$
(24)$$T_{fo}=20\mathrm{lg}\left|T_{D}(j2\pi f_{o})\right|=20\mathrm{lg}\frac{K_{\mathrm{PWM}}(K_{P}+K_{r})}{2\pi f_{o}(L_{1}+L_{2}+L_{g})}$$

*K*_P_ and *K*_r_ can be solved from (23) and (24), whose expressions can be given as follows:(25)$$K_{P}\approx \frac{\omega_{c}(L_{1}+L_{2}+L_{g})}{K_{\mathrm{PWM}}}$$
(26)$$K_{r}=(10^{\frac{T_{fo}}{20}}f_{o}-f_{c})\frac{2\pi (L_{1}+L_{2}+L_{g})}{K_{\mathrm{PWM}}}$$

It is noted that (26) shows the constraint on *K*_r_ for a given *T_fo_.* According to the definition of phase margin PM, the expression of PM can be written as
(27)$$\mathrm{PM}=180^{\circ }+∠T_{D}(j2\pi f_{c})$$

When analyzing the phase frequency characteristic of *T*_D_(s), *G*_PR_(*s*) can be simplified to *K*_P_ + 2 *K*_r_*ω*_i_/*s* at *f*_c_. Substituting *G*_PR_(*s*) = *K*_P_ + 2 *K*_r_*ω*_i_/*s* into (27), (28) is obtained.
(28)$$\begin{array}{l} \mathrm{PM}=\arctan\frac{L_{1}(\omega_{\mathrm{res}}^{2}-\omega_{c}^{2})-K_{\mathrm{PWM}}H\tan(\omega_{c}T_{s}/2)\omega_{c}\cos(1.5\omega_{c}T_{s})}{K_{\mathrm{PWM}}H\tan(\omega_{c}T_{s}/2)\omega_{c}\sin(1.5\omega_{c}T_{s})}\\ \;\;\;\;\;\;-1.5\omega_{c}T_{s}-\arctan\frac{2\pi K_{r}}{\omega_{c}K_{P}} \end{array}$$

By substituting (25) and (26) into (28) and doing some simple mathematical operations, (29) and (30) are obtained.
(29)$$\begin{array}{l} K_{r\_\omega c\_\mathrm{PM}}=\frac{\omega_{c}^{2}(L_{1}+L_{2}+L_{g})}{2\pi K_{\mathrm{PWM}}}\cdot \\ \;\;\;\;\;\frac{(\omega_{r}^{2}-\omega_{c}^{2})L_{1}-HK_{\mathrm{PWM}}\tan(\omega_{c}T_{s}/2)\omega_{c}(\cos(1.5\omega_{c}T)+\tan(\mathrm{PM}+1.5\omega_{c}T_{s})\sin(1.5\omega_{c}T))}{(\omega_{r}^{2}-\omega_{c}^{2})L_{1}\tan(\mathrm{PM}+1.5\omega_{c}T_{s})+HK_{\mathrm{PWM}}\tan(\omega_{c}T_{s}/2)\omega_{c}(\sin(1.5\omega_{c}T)-\tan(\mathrm{PM}+1.5\omega_{c}T_{s})\cos(1.5\omega_{c}T))} \end{array}$$
(30)$$\begin{array}{l} H_{\omega_{c}\_\mathrm{PM}}=\frac{(\omega_{r}^{2}-\omega_{c}^{2})L_{1}}{K_{\mathrm{PWM}}\tan(\omega_{c}T_{s}/2)\omega_{c}}\cdot \\ \;\;\;\frac{\omega_{c}K_{P}-2\pi K_{r}\tan(\mathrm{PM}+1.5\omega_{c}T)}{(\omega_{c}K_{P}-2\pi K_{r}\tan(\mathrm{PM}+1.5\omega_{c}T))\cos(1.5\omega_{c}T)+\sin(1.5\omega_{c}T)(2\pi K_{r}+\omega_{c}K_{P}\tan(\mathrm{PM}+1.5\omega_{c}T))} \end{array}$$

For a given *ω*_c_, (29) shows the constraint on *K*_r_ for a given PM, and for a given *K*_r_, (30) depicts the value range available of *H* and *ω*_c_.

In order to obtain the constraint of the gain margin GM on *H*, the crossing frequencies *f*_1_ and *f*_2_ should be solved first. However, different from the CCFAD whose crossing frequencies are fixed at *f*_s_/6 and *f*_r_, it is hard to obtain the solution of *f*_1_ and *f*_2_ from (31) due to the crossing frequencies *f*_1_ and *f*_2_ for the proposed method varying with *H.* According to the definition of GM, (32) is deduced, which denotes that for given *H*, the corresponding frequency points satisfy the design requirement of GM.
(31)$$H_{\_\omega }=\cos(1.5T_{s}\omega )\frac{(\omega_{r}^{2}-\omega^{2})L_{1}}{K_{\mathrm{PWM}}\tan(T_{s}\omega /2)\omega }$$
(32)$$\begin{array}{l} H_{\omega \_\mathrm{GM}}=\frac{L_{1}}{\omega K_{\mathrm{PWM}}\mathrm{tg}(\omega T_{s}/2)}[(\omega_{r}^{2}-\omega_{}^{2})\cos(1.5\omega T_{s})+\\ \;\;\;\sqrt{{\left(\frac{\omega_{c}\omega_{r}^{2}}{\omega }10^{\frac{GM}{20}}\right)}^{2}-{\left(\omega_{r}^{2}-\omega_{}^{2}\right)}^{2}\sin^{2}(1.5\omega_{}T_{s})}] \end{array}$$

The key to co-designing the current controller and active damper is the determination method of their parameters whose value ranges can be obtained according to the constraint formulas mentioned above and given system stability requirements. A simple determination procedure of the parameters recommended will be given below, which takes the case of *C* = 3.9 μF and the remaining parameters listed in Table 2 as an example.

(1) Give the system stability requirements to ensure system robust stability. It is commonly claimed that PM > π/4 for better dynamic performance and *T_fo_* > 50 dB for lower grid-tied current amplitude error as the grid fundamental frequency fluctuates; besides, for adequate system robustness, GM_1_ = 3 dB at *f*_1_ and GM_2_ = −3 dB at *f*_2_ if *f*_2_ exists.

(2) Identify the resonant frequency *ω*_c_ without taking *L_g_* into account. According to the fact that the higher *f*_c_, the better the system dynamic performance, and considering that *f*_s_ in the experiment in this paper is not high, the value range available of *f*_c_ is the left side of the curve PM = π/4 shown in Figure 10a, and *f*_c_ = 600 Hz is chosen.

(3) After selecting *f*_c_, *K*_p_ is decided by (25) and *K_r_* is chosen in the value range available decided by (26) and (29), which give the lower limit value and upper limit value, respectively. *K*_p_ = 0.05 and *K*_r_ = 75 are selected in this paper.

(4) A bounded range for *H* can be determined by some constraints outlined earlier. Substitute *f*_c_ = 600 Hz and GM_1_ and GM_2_ with different values into (32) to get different curves, as shown in Figure 10b. The curves in Figure 10b are divided into left and right parts according to frequency range. In the left half plane of Figure 10b, the curve *H_ω* and two *H_*_GM_ curves representing GM_1_ = 3 dB and GM_1_ = 3.3 dB overlap after point *a* and point *b* from left to right, respectively, where the overlapping part means that the two curves *H_ω* and *H_*_GM_ have common solutions of *H* and *f*_1_ on the condition that the requirement for GM_1_ is met, and it can be seen that the intersection point moves to the right when the value of GM_1_ increases, which indicates that the range value available of *H* gets smaller accordingly. Therefore, the *H_*_GM1_ corresponding to point *a* is the maximum value in the case of GM_1_ ≥ 3 dB. In the right half plane of Figure 10b, the curve *H_ω* and two *H_*_GM_ curves representing GM_2_ = −3 dB and GM_1_ = −5 dB, respectively, overlap after point *c* and point *d* from left to right, respectively, where the overlapping part means that the two curves *H_ω* and *H_*_GM_ have common solutions of *H* and *f*_2_ on the condition that the requirement for GM_2_ is met, and the intersection point moves to the right when the value of GM_2_ increases. Therefore, the *H* corresponding to point *e* is the maximum value in the case of GM_2_ ≥ −3 dB. Based on the above analysis, obviously, under the condition that the requirements for GM_1_ and GM_2_ are met simultaneously, the value range of *H* is [*H*_D_*, H___*_GM2_], in which the frequency points *f*_1_, *f*_2_, and *f*_3_ corresponding to any *H*_ran_ are consistent with that shown in the Figure 9c.

(5) Except for the value range limitations mentioned in (4), the determination of the best possible damping gain *H* requires one more consideration, which is to place the two resonant poles of system as far as possible inside the unit circle to achieve the maximum damping. A pole-zero map strategy is used to determine the selection of *H* in Figure 11a. Based on all consideration above, for the example system, a better choice of *H* occurs when *H* = 0.04 marked in Figure 11a.

(6) Figure 11b demonstrates the feasibility of the choice of *H* when *L_g_* changes from 0 to maximum value (0.1 pu). It is clear that the resonant poles are always inside the unit circle, which indicates that the system stability margin can be guaranteed all the time. Otherwise, return to (2) and reselect another *f*_c_.

Based on the above analysis, the flowchart of the design procedure can be drawn as shown in Figure 12.

## 6. Verification

In order to verify the correctness of the theoretical analysis in Section 4 and Section 5, simulations and experiments are conducted with the parameters listed in Table 4.

### 6.1. Simulation Results

A simulation model of a three-phase LCL grid-connected inverter system was built using Matlab/Simulink simulation platform, which was conducted with the parameters listed in Table 4.

The impedance of the power grid consists of resistance and inductance when the system is in the medium- and low-voltage distribution networks. Using the proposed method, choose H = 0.1 while keeping the power grid inductance *L*_g_ = 2.0 mH unchanged, and run simulations with varied power grid resistance values, i.e., varying reactance to resistance ratios *X*_g_/*R*_g_. According to the simulation results, the grid resistance *R* progressively increases from zero, and the resonance and THD of the grid current change. When *R* = 0 Ω, the grid current experiences a slight resonance, the high-frequency signal is amplified, and the THD increases. Because of the damping effect of resistance, as *R* grows, the grid current’s resonance reduces gradually, and the THD steadily falls. As *R* grows further, the content of the grid current low-order harmonics, notably the 5th and 7th harmonics, increases, causing a gradual rise in the THD. As a result, the system may become unstable. To ensure system stability in this instance, extra hardware changes and software control are required to obtain a suitable *X*_g_/*R*_g_. Figure 13a,b, respectively, provide simulation results and THD analysis results of grid current waveform with different R values, namely 0 Ω, 3 Ω, and 6 Ω. In order to better verify the harmonic suppression ability of the algorithm, the worst-case conditions are considered, assuming that the grid impedance is purely inductive, i.e., *R* = 0 Ω.

In addition, according to the analysis in the previous section, using the proposed method, H = 0.4 is selected for simulation under different grid inductance conditions. Figure 13c shows the simulation results under *L_g_* = 2.0 mH, *L_g_* = 10 mH, and *L_g_* = 20 mH, which correspond to SCR =10, SCR =2, and SCR =1, respectively. As can be seen, the grid current is stable in three cases and the results is consistent with Figure 12b, which means that the design procedure is effective and indicates that the system can stably operate in ultraweak grid.

The simulation is conducted with the controller parameters *K*_p_ = 0.05 and *K*_r_ = 75. Figure 14 shows the comparison results between the proposed method and CCFAD. As can be seen, in the case of *L_g_ =* 2.0 mH and *C* = 9.4 μF, the filter resonant frequency *f*_r_ = 1.49 KHz is smaller than *f*_s_/6, which is within the EDR of the two methods. By setting the controller parameters reasonably, resonance will not occur in the grid current. As shown in Figure 14a, after using the proposed method, there was no significant change in the grid current and the system was in a stable state. In the case of *L_g_ =* 0.8 mH and *C* = 9.4 μF, the filter resonant frequency *f*_r_ = 1.68 KHz is approximately equal to *f*_s_/6. At this time, no matter how to adjust the control parameters, CCFAD cannot meet the requirements of two magnitude margins at the same time, so the system is in a marginal stability state. However, the resonance frequency *f*_r_ is within the EDR range when using the proposed method. To ensure system stability while designing parameters, just the requirement for GM_1_ must be satisfied. As shown in Figure 14b, there is a slight resonance in the grid current on the left, but after adopting the proposed method, the resonance in the grid current on the right disappears. In the case of *C* = 3.9 μF and *L_g_* = 0 mH, the filter resonant frequency *f*_r_ = 3.12 KHz is slightly less than *f*_s_/3, so it is in the EDR of the proposed method. When the proportional feedback coefficient *H* of the damping loop is changed from 0.12 to 0.04, according to Figure 12a, it can be seen that the positions of the two resonant poles of the system enter the unit circle from outside, indicating that the system transitions from unstable to stable. Figure 14c shows that the resonance in the grid-connected current disappears, and the simulation results agree with that depicted in Figure 12b.

### 6.2. Experimental Results

An experimental platform of 6 kVA three-phase, grid-tied inverter is built, for which the system parameters are shown in Table 3. Figure 15 shows the prototype photograph. We use TMS320F28035 chip of TI company to process digital signals, and use INFINEON’s FF50R12RT4 IGBT as the power switch. Also, a programmable AC source (TPV7006S) series inductance is used to simulate the actual power grid. The steady state and load jump test waveforms of grid-tied inverter are given below.

The experiments adopt the same controller parameters as the simulation. For the convenience of direct comparison, the experiments in Figure 16a,b are conducted by switching between the two methods, and those in Figure 16c are done by manually changing from 0.12 to 0.04 in the parameter H using the proposed method. It can be observed that the experimental waveforms are basically consistent with the simulation results. Besides, it should be pointed out that each individual harmonic distortion and the measured total harmonic distortion (THD) shown in Figure 16d all meet the distortion limits in IEEE Standard 519-2014 [41].

Figure 17 shows the experimental results of phase-A with the proposed method in the case of the *L_g_ =* 0 mH and *L_g_ =* 2.0 mH. The experimental result agrees with that depicted in Figure 12b. Combining the experimental results in Figure 16, it can be summarized that the system using the proposed method can maintain stable operation and exhibit a good damping effect under the condition that the grid impedance is within the range of 0~2 mH, indicating that the system has good robustness to the grid impedance.

Finally, Figure 18 shows the transient experimental results when the current reference changes based on different *L*_g_. The instantaneous overshoot and adjustment time are both very small, indicating that the new method has good dynamic performance.

## 7. Conclusions

For the digital control LCL-type grid-tie inverter system, its digital control delay will reduce the system’s robustness to the grid impedance, which may lead to system instability. The article proposes an improved digital differentiator, which can be proven to be a better way to achieve an ideal differentiator. Compared with the ideal differentiator, it has identical phase frequency characteristics that can ensure that the system EDR reaches [0, *f*_s_/3] and has better noise rejection capability. In addition, due to their direct discrete nature, compact expressions, and simple algebraic operations, they are more attractive. The tested results from the experiments verified that the proposed active damping method maintains stable operation in weak grids, obtaining high robustness against grid impedance variation. The proposed method emulates the ideal differentiator and increases EDR, which is still relatively small. Future research will focus on further improving EDR.

## Figures and Tables

**Figure 1 sensors-23-08203-f001:**
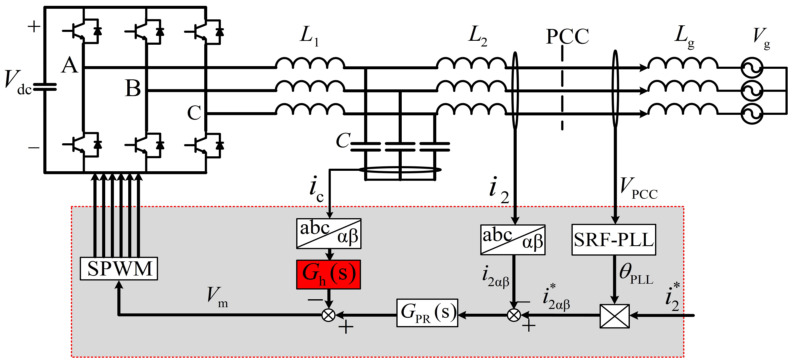
Topology and current control architecture of a three-phase, LCL-type, grid-tied inverter.

**Figure 2 sensors-23-08203-f002:**
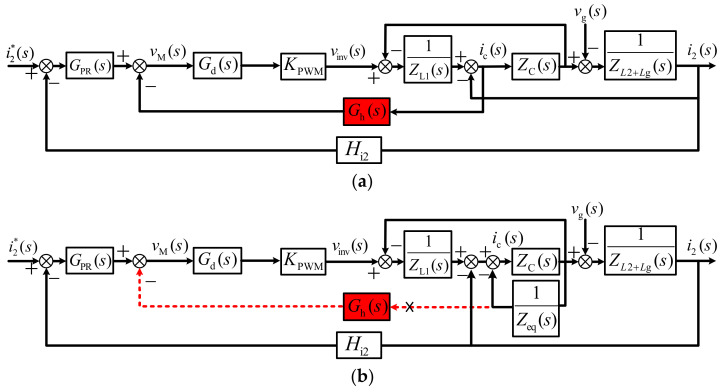
Block diagrams of inverter with CCF active damping. (**a**) Initial one. (**b**) Equivalent one.

**Figure 3 sensors-23-08203-f003:**
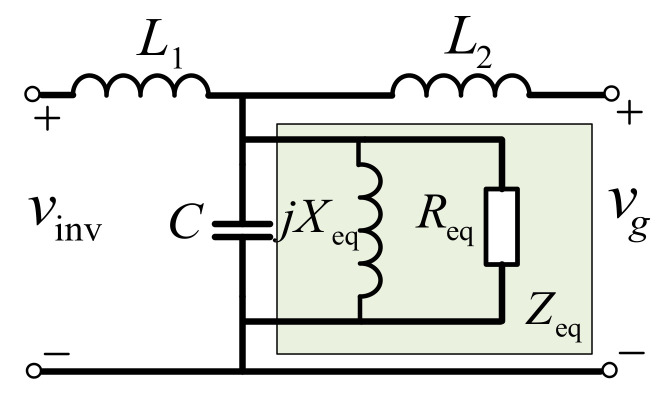
The equivalent circuits of virtual impedance.

**Figure 4 sensors-23-08203-f004:**
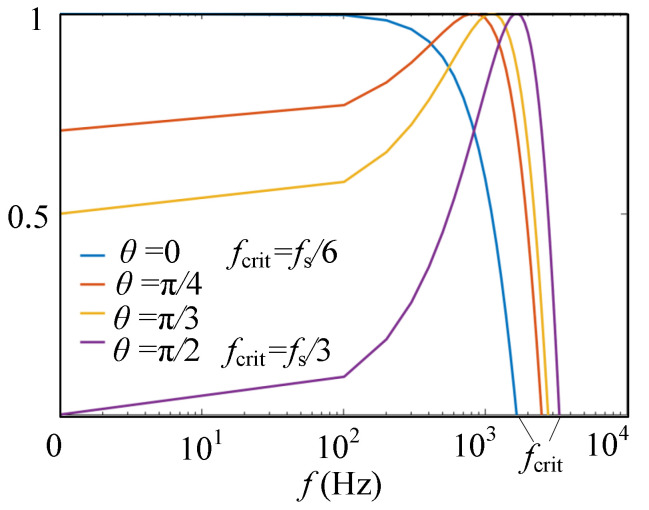
Positive region of equivalent resistance *R*_eq_(*ω*).

**Figure 5 sensors-23-08203-f005:**
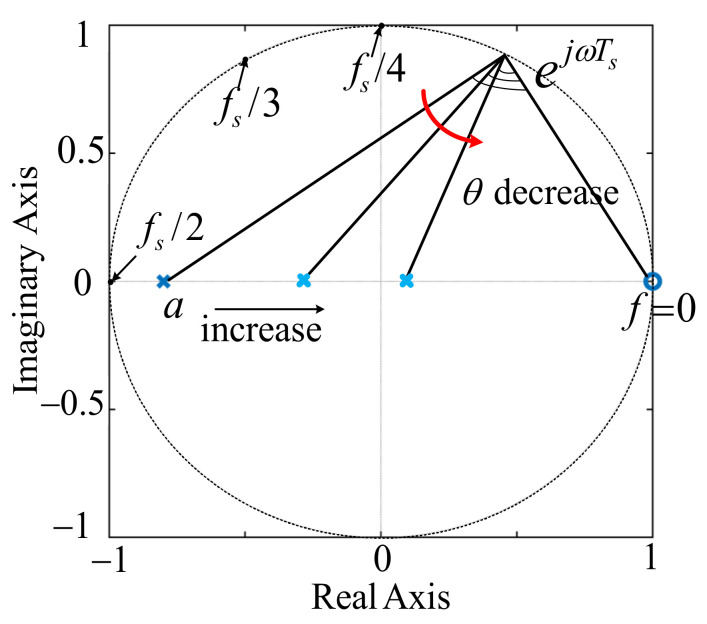
Zero-pole distribution of *G*_h_(*s*).

**Figure 6 sensors-23-08203-f006:**
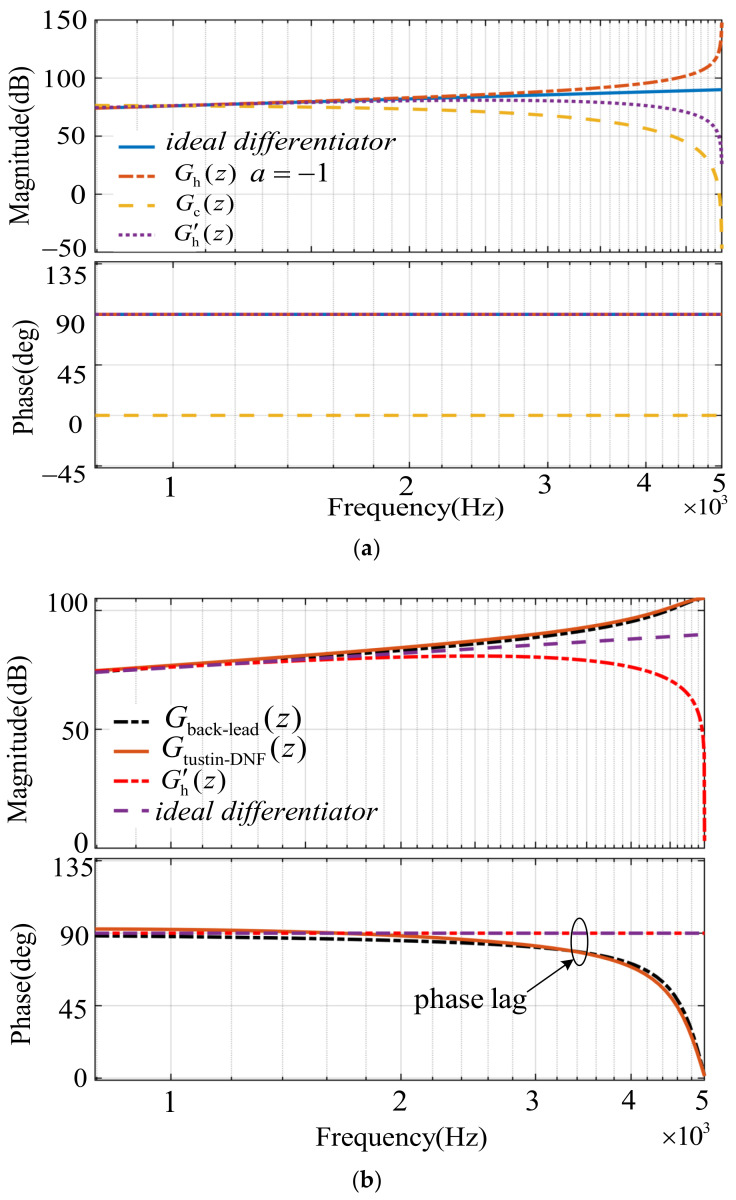
Frequency response of various functions. (**a**) Frequency response of *G*_h_(*z*), *G*_c_(*z*), $G_{h}^{\prime }(z)$, and the ideal differentiator. (**b**) Frequency response of *G*_back-lead_(*z*), *G*_tustin-DNF_(*z*), *G*_c_(*z*), $G_{h}^{\prime }(z)$, and the ideal differentiator.

**Figure 7 sensors-23-08203-f007:**
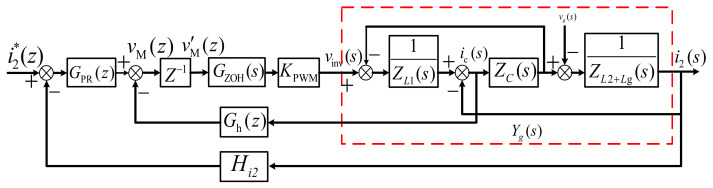
Block diagram of CCFAD in the discrete domain.

**Figure 8 sensors-23-08203-f008:**
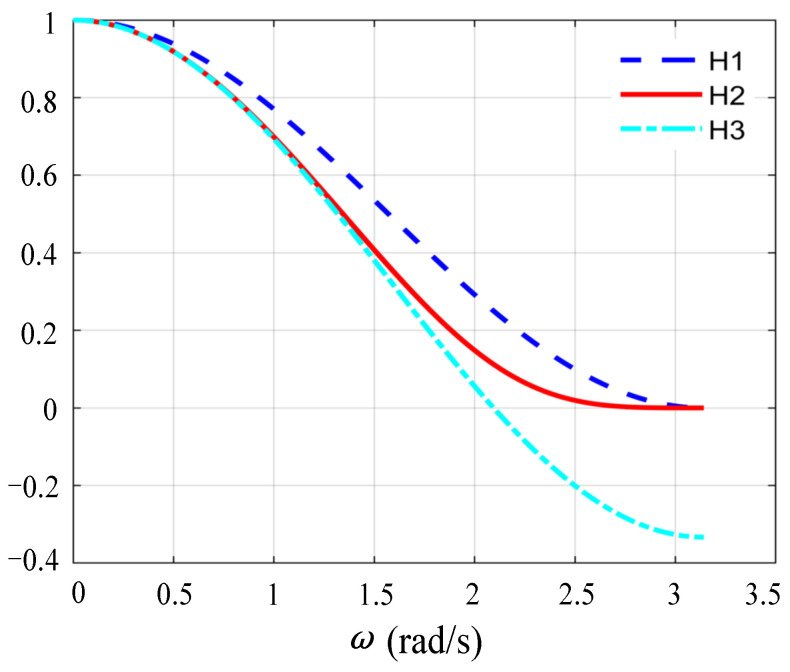
Comparison of *H*_1_, *H*_2_, and *H*_3_.

**Figure 9 sensors-23-08203-f009:**
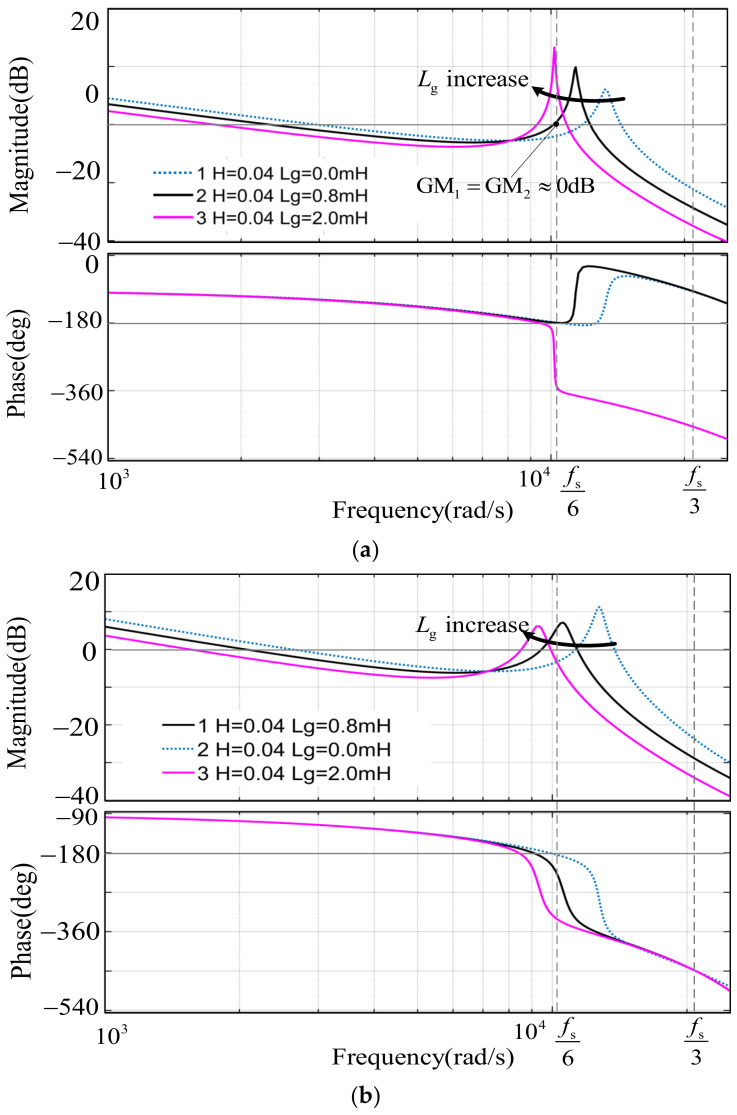
Bode diagrams of the open-loop transfer function *T*_D_(*z*) with *G*_PR_(z) = 1. (**a**) *G*_h_(z) = *H*, *C* = 9.4 µF; (**b**) *G*_h_(*z*) = *H*(*z* − 1)/(*z* + 1), *C* = 9.4 µF; (**c**) *G*_h_(*z*) = *H*(*z* − 1)/(*z* + 1), *C* = 3.9 µF.

**Figure 10 sensors-23-08203-f010:**
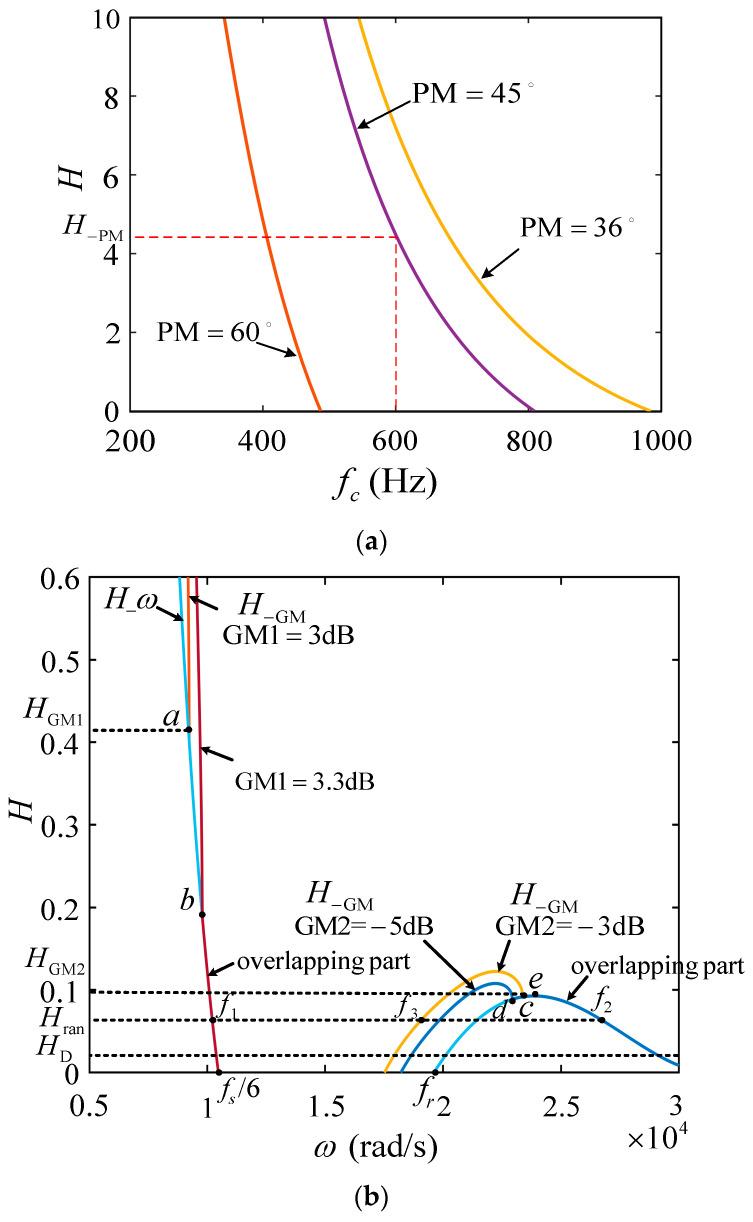
(**a**) Value range available of *f*_c_ and *H* under the constraint of PM. (**b**) Value range available of *ω*_c_ and *H* under the constraint of GM_1_ and GM_2_.

**Figure 11 sensors-23-08203-f011:**
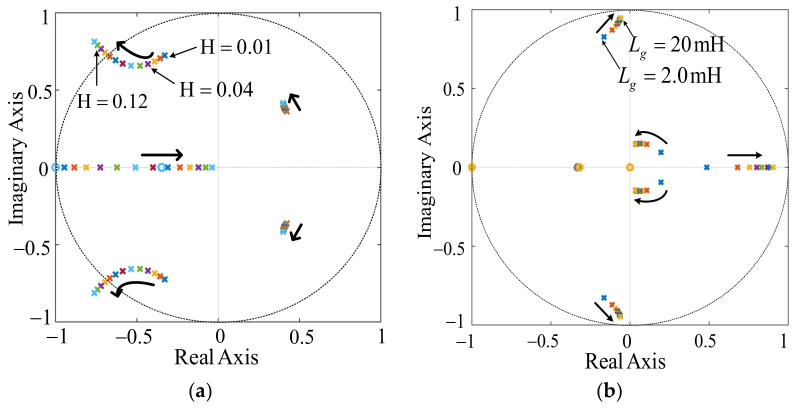
Pole-zero map of system. (**a**) *L_g_* = 0 for different values of *H.* (**b**) different values of *L_g_* for given *H* = 0.04.

**Figure 12 sensors-23-08203-f012:**
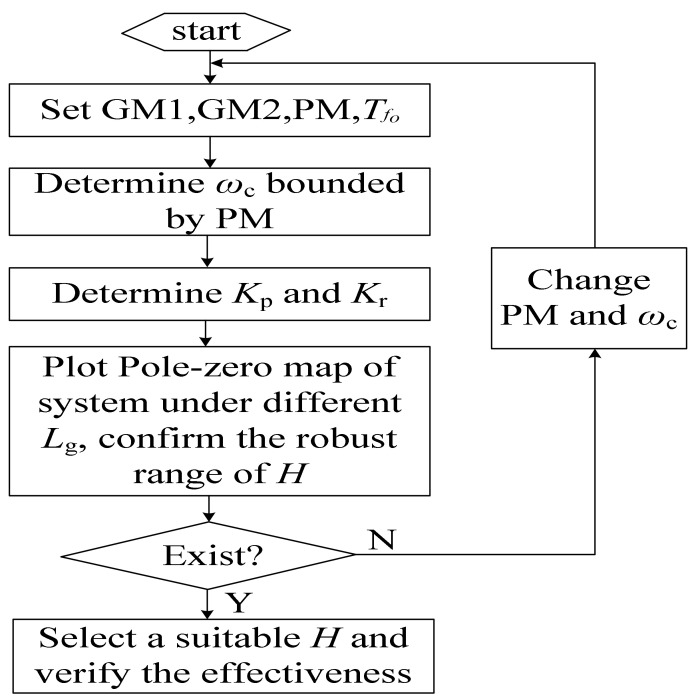
Flowchart of the design procedure.

**Figure 13 sensors-23-08203-f013:**
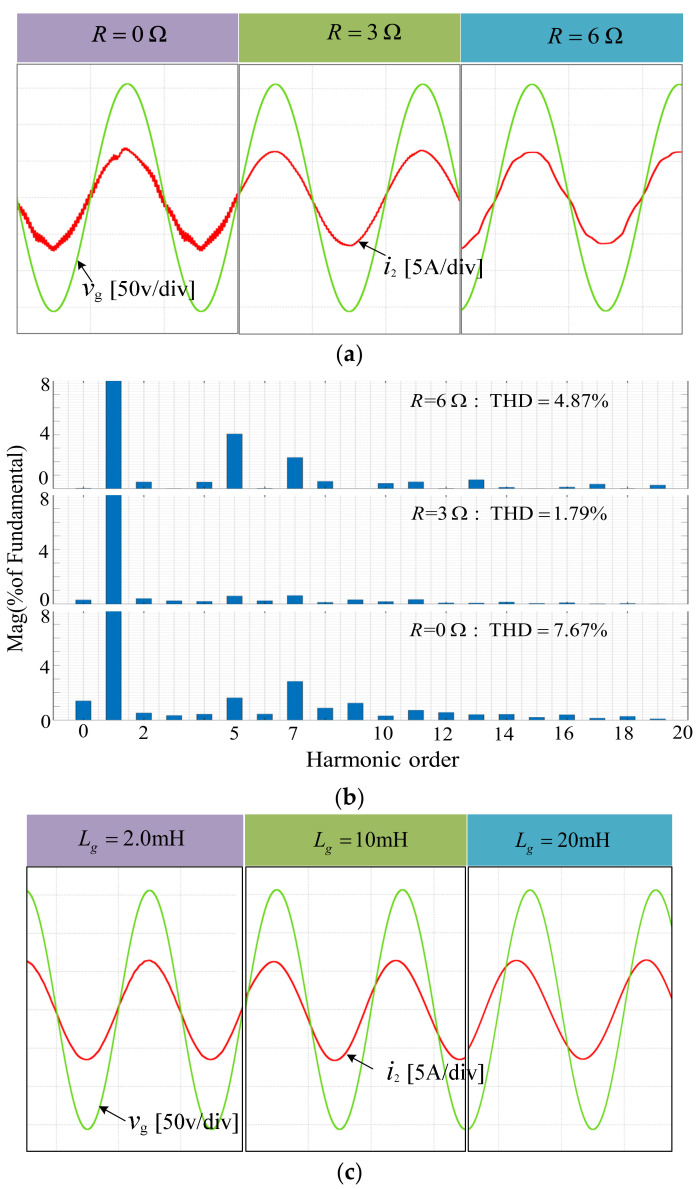
The simulation results of improved CCFAD (**a**) the simulation results under different *R.* (**b**) Spectra of grid-connected current under 3 situations above. (**c**) the simulation results under *L_g_* = 2.0 mH, *L_g_* = 10 mH and *L_g_* = 20 mH.

**Figure 14 sensors-23-08203-f014:**
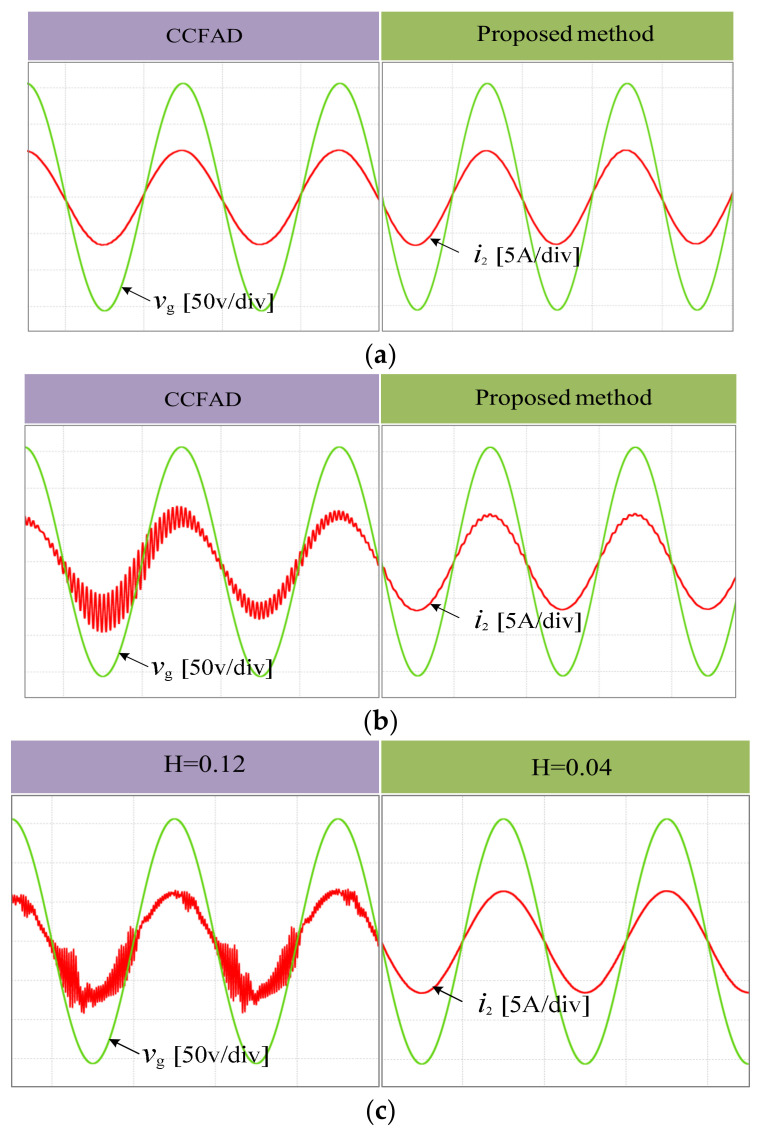
Transient simulation results of improved CCFAD (**a**) *L_g_ =* 2.0 mH, *C* = 9.4 μF. (**b**) *L_g_ =* 0.8 mH, *C* = 9.4 μF. (**c**) *L_g_ =* 0 mH, *C* = 3.9 μF.

**Figure 15 sensors-23-08203-f015:**
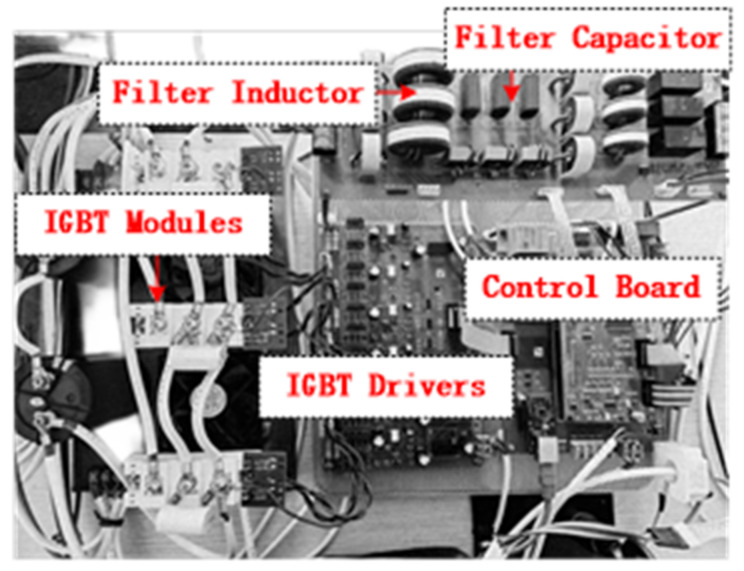
Prototype photograph.

**Figure 16 sensors-23-08203-f016:**
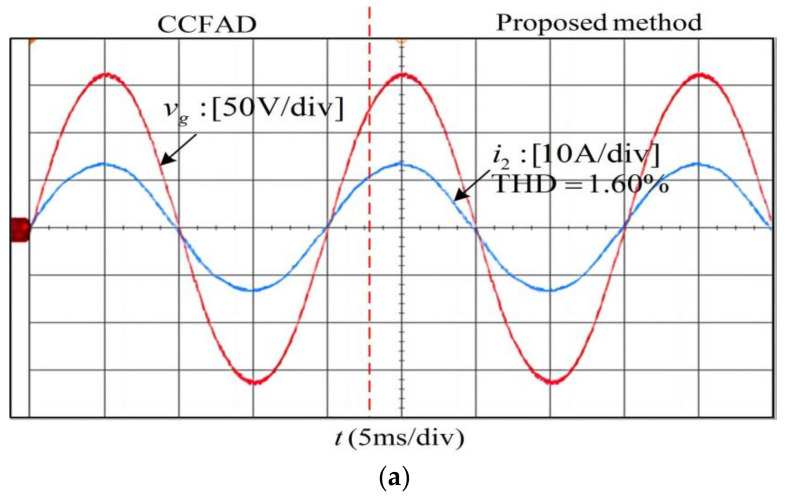
Transient experimental results of improved CCFAD (**a**) *L_g_ =* 2.0 mH*, C* = 9.4 μF. (**b**) *L_g_ =* 0.8 mH, *C* = 9.4 μF. (**c**) *L_g_ =* 0 mH, *C* = 3.9 μF. (**d**) Spectra of grid-connected current under 3 situations above.

**Figure 17 sensors-23-08203-f017:**
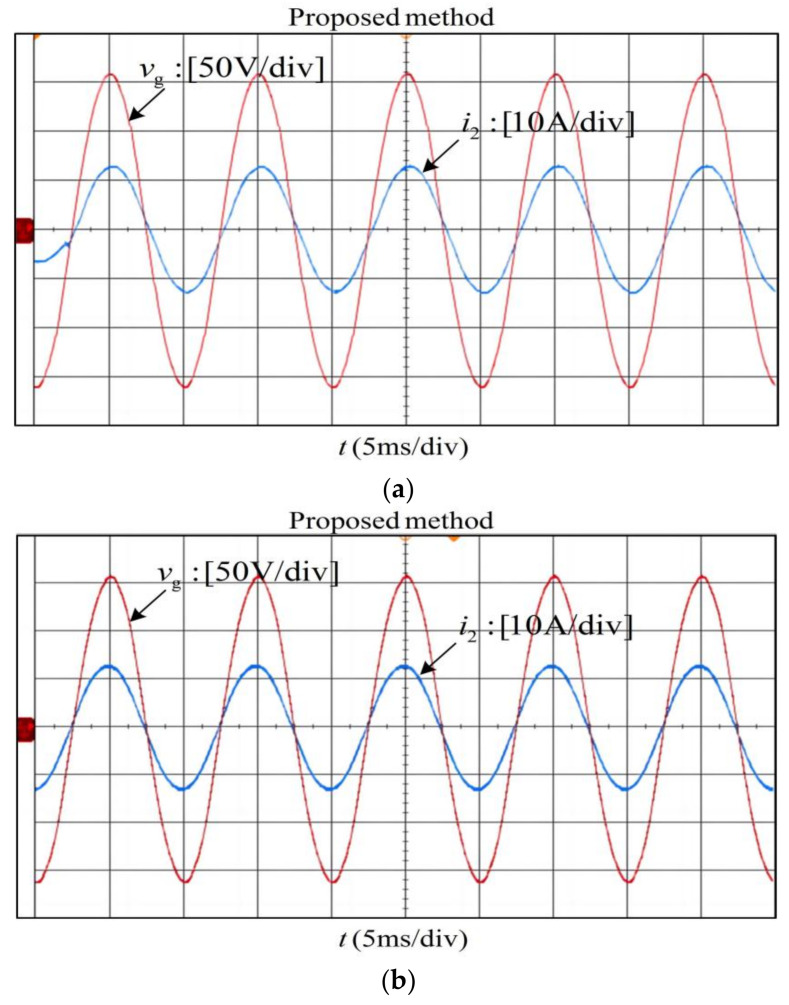
Experimental results of phase-A with the proposed method when *L*_g_ varies. (**a**) *C* = 3.9 μF, *L*_g_ = 0 mH. (**b**) *C* = 3.9 μF, *L*_g_ = 2.0 mH.

**Figure 18 sensors-23-08203-f018:**
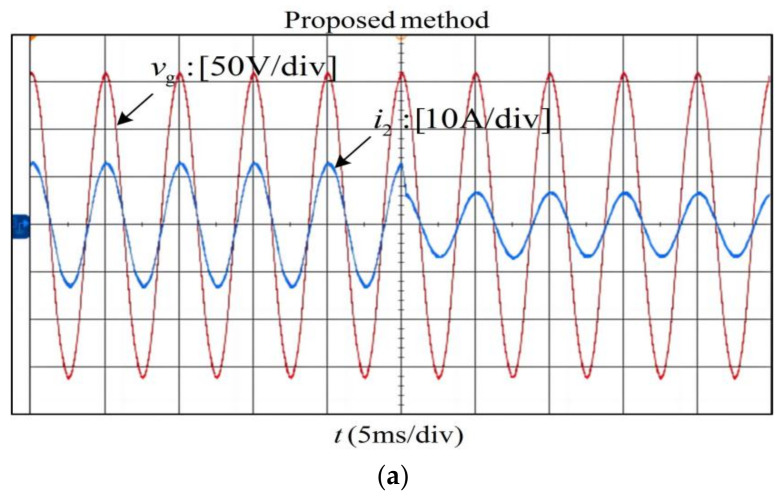
Transient experimental results when the current reference changes based on different *L*_g_. (**a**) *L*_g_ = 0 mH. (**b**) *L*_g_ = 2.0 mH.

**Table 1 sensors-23-08203-t001:** Comparison of different digital compensation methods.

Features	Lead Link in [17]	First-Order High-Pass Filter	Second-Order High-Pass Filter	Backward-Lead	Tustin-DNF	Proposed Method
Function order	First order	First order	First order	First order	First order	First order
Simplicity	good	better	inferior	better	good	better
Effective frequency range	[0, *f*_s_/4]	[0, *f*_s_/3)	[0, *f*_s_/2)	[0, *f*_s_/3)	[0, *f*_s_/3)	[0, *f*_s_/3]

**Table 2 sensors-23-08203-t002:** Results of the Routh criterion.

Case	Requirement	*a* _1_	*b* _1_	*c* _1_	RHP Poles
1	*H* < *H*_3_	+	+	+	0
2	*H*_3_ < *H*< *H*_2_	+	+	−	2
3	*H*_2_ < *H* < *H*_1_	+	−	+	2
4	*H* > *H*_1_	−	+	+	2

**Table 3 sensors-23-08203-t003:** Stability requirement comparison of digitally controlled inverter with different active damping methods.

Case	Method	$f_{r}$	*f*_r_′	*R*_eq_(*f*_r_′)	*P*	−180°-Crossing	Magnitude Margin (GM) Requirement	Phase Margin (PM) Requirement
1	1	(0, *f*_s_/6)	(0, *f*_s_/6)	+	0	*f*_r_-	GM_1_ > 0 dB	PM > 0°
2			[*f*_s_/6, *f*_s_/2)	−	2	*f*_r_−, *f*_s_/6+	GM_1_ > 0 dB, GM_2_ < 0 dB	PM > 0°
3		*f*_s_/6	(*f*_s_/6, *f*_s_/2)	−	2	None	Unstable	Unstable
4		(*f*_s_/6, *f*_s_/2)	(*f*_s_/6, *f*_s_/2)	−	2	*f*_s_/6−, *f*_r_+	GM_1_ < 0 dB, GM_2_ > 0 dB	PM > 0°
1	2	(0, *f*_s_/3)	(0, *f*_s_/3)	+	0	*f*_1_−	GM_1_ > 0 dB	PM > 0°
2			[*f*_s_/3, *f*_s_/2)	−	2	*f*_1_−, *f*_2_+	GM_1_ > 0 dB, GM_2_ < 0 dB	PM > 0°
3		[*f*_s_/3, *f*_s_/2)	[*f*_s_/3, *f*_s_/2)	−	2	*f*_1_−, *f*_2_+	GM_1_ > 0 dB, GM_2_ < 0 dB	PM > 0°

**Table 4 sensors-23-08203-t004:** Main circuit parameters.

Symbol	Parameter Name	Value
*V* _o_	Inverter output phase voltage	110 V
*f* _o_	Fundamental frequency	50 Hz
*f* _sw_	Switching frequency	10 kHz
*f* _s_	Sampling frequency	10 kHz
*T* * _s_ *	Sample period	100 μs
*V* _dc_	DC-link voltage	400 V
*L* _1_	Converter-side filter inductance	2.0 mH
*L* _2_	Grid-side filter inductance	1.0 mH
*C*	Filter capacitor	9.4/3.9 μF
*L* * _g_ *	Grid inductance	0~2 mH (0.1 *pu*)
*K* _PWM_	Equivalent gain of inverter	60
*H* * _i_ * _2_	Grid current sample coefficient	1

## Data Availability

Data is contained within this article.

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
