# Peer review of "An Improved Active Damping Method for Enhancing Robustness of LCL-Type, Grid-Tied Inverters under Weak Grid Conditions"

_sensors, 2023, doi:10.3390/s23198203_

Round 1
Reviewer 1 Report
Authors can improve the quality of this article by addressing the following comments:
1- Authors should provide suitable references for equations.
2- Please add a comparative table to compare the proposed methodology with recent works in the same field.
3- Does the proposed approach guarantee the converging to the global optimal or close to the global optimal?
4- Introduction section must be enhanced by discussing some recent works such as: A three-layer game theoretic-based strategy for optimal scheduling of microgrids by leveraging a dynamic demand response program designer to unlock the potential of smart buildings and electric vehicle fleets; Improved double‐surface sliding mode observer for flux and speed estimation of induction motors; A tri-layer stochastic framework to manage electricity market within a smart community in the presence of energy storage systems; An interval-based nested optimization framework for deriving flexibility from smart buildings and electric vehicle fleets in the TSO-DSO coordination;
5- The information on the computer used for simulation must be added to the paper.
6- Please provide more descriptions for formulations.
7- Please enhance the conclusion section by adding some numerical results to it.
Author Response
I sincerely appreciate your comments, which have been very beneficial to me, and I hope my reply can meet your requirements.
Reviewer: 1
- Authors should provide suitable references for equations.
Reply: Added reference literature to the cited equations. referring to the equations.
- Please add a comparative table to compare the proposed methodology with recent works in the same field.
Reply: A comparison table was added to compare the proposed method with recent works in the same field, referring to Table 1 in page 8.
- Does the proposed approach guarantee the converging to the global optimal or close to the global optimal?
Reply: The proposed method has been analyzed using the Routh criterion and zero pole analysis to obtain stable conditions. I am sorry, I am not sure how to demonstrate whether the proposed method can converge to the global or close to the global optimal.
- Introduction section must be enhanced by discussing some recent works such as: A three-layer game theoretic-based strategy for optimal scheduling of microgrids by leveraging a dynamic demand response program designer to unlock the potential of smart buildings and electric vehicle fleets; Improved double‐surface sliding mode observer for flux and speed estimation of induction motors; A tri-layer stochastic framework to manage electricity market within a smart community in the presence of energy storage systems; An interval-based nested optimization framework for deriving flexibility from smart buildings and electric vehicle fleets in the TSO-DSO coordination;
Reply: The introduction was rewritten and several recent literature was added into the comparative discussions. The revised content is highlighted in blue in the newly provided manuscript in page 2.
- The information on the computer used for simulation must be added to the paper.
Reply: The verification section added simulation results, and related information on the computer used.The revised content is highlighted in blue in the newly provided manuscript in page 19.
- Please provide more descriptions for formulations.
Reply: The contribution and novelty of the article are re presented at the end of the introduction. The revised content is highlighted in blue in the newly provided manuscript in page 3.
- Please enhance the conclusion section by adding some numerical results to it.
Reply: The conclusion section was rewritten. The revised content is highlighted in blue in the newly provided manuscript in page 22.
Reviewer 2 Report
1. The similarity index is %27 with references. It must be less than %20 with references.
2. More references must be given in introduction part comparatively and the novelty must be given openly.
3. The size of the figure 1 is very small. It must be enlarged to be read all the text.
4. How do you derive Equation 1?
5. Figure 3 must be enlarged to be read in %100.
6. The figures must be located correct areas in the text. The ex must be before the figures.
7. Fig 3 is which part of the main topology. It must be clarifying.
8. All the parameters in equation can be given in nomenclature part.
9. What is the y axıs of figure 4?
10. Figure 6 and 7 commended more detailed.
11. The equation derivation part must be more detailed. They can be given in Appendix.
12. The Routh criteria analyses (Eq19-21) include more details they can be given in appendix.
13. In Table 3. Main Circuit Parameters are which circuit parameters. İt is not clear.
14. Figure 9 must be enlarged and commend more detailed.
15. The typo errors in Eq. 32 and 33 must be corrected.
16. There is only phase gain and margin analyses. Where is Nyquist and root analyses?
17. Figure 12 and 13 must be given in a comparable form (a) (b)..
18. Figure 1 must be colour and single line diagram of the circuit must be given to be understand more easily the setup.
19. Some abbreviation must be given openly.
20. There are only experimental results. They can be compared with simulations. Also a comparison table for the results.
21. The conclusion part can be more detailed.
22. The references must be increased. It is very less for a comprehensive study. Some of the useful paper to be read;
Xu, J., Xie, S., Zhang, B., & Qian, Q. (2018). Robust grid current control with impedance-phase shaping for LCL-filtered inverters in weak and distorted grid. IEEE Transactions on Power Electronics, 33(12), 10240-10250.
Li, X., Wu, X., Geng, Y., Yuan, X., Xia, C., & Zhang, X. (2014). Wide damping region for LCL-type grid-connected inverter with an improved capacitor-current-feedback method. IEEE Transactions on Power Electronics, 30(9), 5247-5259.
Şahin, M. E., & Okumuş, H. İ. (2018). Comparison of different controllers and stability analysis for photovoltaic powered buck-boost DC-DC converter. Electric Power Components and Systems, 46(2), 149-161.
Pena-Alzola, R., Liserre, M., Blaabjerg, F., Ordonez, M., & Yang, Y. (2014). LCL-filter design for robust active damping in grid-connected converters. IEEE Transactions on Industrial Informatics, 10(4), 2192-2203.
Fu, X., & Li, S. (2015). Control of single-phase grid-connected converters with LCL filters using recurrent neural network and conventional control methods. IEEE Transactions on Power Electronics, 31(7), 5354-5364.
Sahin, M. E., & Okumus, H. I. (2013, October). Small signal analyses and hardware implementation of a buck-boost converter for renewable energy applications. In 2013 ICRERA (pp. 330-335). IEEE.
23. What is the contribution of the authors and any Project support for his paper: It must be given?
24. The typo and grammar errors must be checked.
25. Generally, the schema of the manuscript must be revised comprehensively to increase the paper fluently track.
Author Response
I sincerely appreciate your comments, which have been very beneficial to me, and I hope my reply can meet your requirements.
Reviewer: 2
- The similarity index is %27 with references. It must be less than %20 with references.
Reply: The introduction was rewritten and several recent literature was added into the comparative discussions. The revised content is highlighted in blue in the newly provided manuscript in page 2.
- More references must be given in introduction part comparatively and the novelty must be given openly.
Reply: 7-8 references were added into the introduction section. And the contribution and novelty of the article are re presented at the end of the introduction. The revised content is highlighted in blue in the newly provided manuscript in page 2,3.
- The size of the figure 1 is very small. It must be enlarged to be read all the text.
Reply:Figure 1 was enlarged and it is clear enough to read. Referring to the Fig.1 in page 3.
- How do you derive Equation 1?
Reply:the references was added to Equation 1. referring to the page 3.
- Figure 3 must be enlarged to be read in %100.
Reply:Figure 3 was enlarged and it is clear enough to read. Referring to the Fig.3 in page 5.
- The figures must be located correct areas in the text. The text must be before the figures.
Reply:The position of Figure 3 has been adjusted after the text. Referring to the Fig.3 in page 5.
- Fig 3 is which part of the main topology. It must be clarifying.
Reply:Figure 3 shows the equivalent impedance Zeq in Figure 2b, as explained in the text.
- All the parameters in equation can be given in nomenclature part.
Reply:All parameters in the equation are introduced in the following text.
- What is the y axıs of figure 4?
Reply:Y axis of Figure 4 has no dimensions, and its focus is on highlighting the positive and negative region of Req (Req is a cosine function, and its magnitude is [-1,1])
- Figure 6 and 7 commended more detailed.
Reply:Figure 6 has added a detailed description, Figure 7 has been enlarged, and a comparative table has been added. Referring to the Figure 6 , 7 and Table 1 in the page 8.
- The equation derivation part must be more detailed. They can be given in Appendix.
Reply:The derivation process of the equation only involves simple mathematical transformations, and I think that directly providing the results is more intuitive.
- The Routh criteria analyses (Eq19-21) include more details they can be given in appendix.
Reply:The Routh criteria analyses (Eq19-21) were placed in appendix. Referring to page 24.
- In Table 3. Main Circuit Parameters are which circuit parameters. İt is not clear.
Replay:The main circuit parameters include L1 L2 Lg C Vdc, while the others are controller parameters.
- Figure 9 must be enlarged and commend more detailed.
Reply: Three images were enlarged and partially modified. Referring to Fig.9 in page 14.
- The typo errors in Eq. 32 and 33 must be corrected.
Reply:The typo errors in Eq. 32 and 33 were corrected, referring to Eq.29 and 30 in page 16.
- There is only phase gain and margin analyses. Where is Nyquist and root analyses?
Reply:In addition to the analysis of the Bode plot, there is also the analysis of the zero and pole points, as shown in Figure 12.
- Figure 12 and 13 must be given in a comparable form (a) (b)..
Reply:The two images have been merged, and Figure 12 and 13 were given in a comparable form, i.e.,Fig. 12(a) and Fig. (b),respectively. Referring to Figure 12 in page 18.
- Figure 1 must be colour and single line diagram of the circuit must be given to be understand more easily the setup.
Reply: Figure 1 was enlarged and easy to read. Figure 1 is colour and a common Topology.
- Some abbreviation must be given openly.
Reply: All abbreviation were given openly at the end of the paper. Referring to the page 24.
- There are only experimental results. They can be compared with simulations. Also a comparison table for the results.
Reply: Some simulation results and discussions have been included in the experimental verification section. The revised content is highlighted in blue in the newly provided manuscript in page 19.
- The conclusion part can be more detailed.
Reply:The conclusion section was rewritten. The revised content is highlighted in blue in the newly provided manuscript in page 22.
- The references must be increased. It is very less for a comprehensive study. Some of the useful paper to be read;
Reply: 7-8 references were added into the introduction section. And the contribution and novelty of the article are re presented at the end of the introduction, including the following recommended articles. The revised content is highlighted in blue in the newly provided manuscript in page 2,3.
Xu, J., Xie, S., Zhang, B., & Qian, Q. (2018). Robust grid current control with impedance-phase shaping for LCL-filtered inverters in weak and distorted grid. IEEE Transactions on Power Electronics, 33(12), 10240-10250.
Li, X., Wu, X., Geng, Y., Yuan, X., Xia, C., & Zhang, X. (2014). Wide damping region for LCL-type grid-connected inverter with an improved capacitor-current-feedback method. IEEE Transactions on Power Electronics, 30(9), 5247-5259.
Şahin, M. E., & Okumuş, H. İ. (2018). Comparison of different controllers and stability analysis for photovoltaic powered buck-boost DC-DC converter. Electric Power Components and Systems, 46(2), 149-161.
Pena-Alzola, R., Liserre, M., Blaabjerg, F., Ordonez, M., & Yang, Y. (2014). LCL-filter design for robust active damping in grid-connected converters. IEEE Transactions on Industrial Informatics, 10(4), 2192-2203.
Fu, X., & Li, S. (2015). Control of single-phase grid-connected converters with LCL filters using recurrent neural network and conventional control methods. IEEE Transactions on Power Electronics, 31(7), 5354-5364.
Sahin, M. E., & Okumus, H. I. (2013, October). Small signal analyses and hardware implementation of a buck-boost converter for renewable energy applications. In 2013 ICRERA (pp. 330-335). IEEE.
- What is the contribution of the authors and any Project support for his paper: It must be given?
Reply:When submitting the manuscript, relevant information was already provided to the editor.
- The typo and grammar errors must be checked.
Reply:I have reviewed the entire article and corrected some errors. If the article is accepted, I plan to use the English polishing service recommended by the journal to improve the article
- Generally, the schema of the manuscript must be revised comprehensively to increase the paper fluently track.
Reply:I have reviewed the entire article and corrected some errors. If the article is accepted, I plan to use the English polishing service recommended by the journal to improve the article
Reviewer 3 Report
The submitted paper presents an improved approach for the active damping of LCL-based grid-connected inverters, relying on the Tustin discrete approach. I have the following comments:
1) The literature can be further improved by refining the research gap and emphasizing the motivation. Additionally, consider providing a comparison table that outlines and compares these techniques.
2) Create a flowchart to summarize the proposed approach.
3) Although the paper already presents experimental time-domain results, it might be insightful to include simulation results that explore various scenarios, such as different SCR and Xg/Rg ratios.
4) Provide further elaboration on the mechanism employed (whether it's automatic or manual) for the practical real-time implementation depicted in Figure 15. Additionally, detail the process by which the transition from the value of H from 0.12 to 0.04 is accomplished.
5) For Figure 4, it would be beneficial to use a logarithmic scale. Moreover, for effective illustration, please include both the magnitude and phase in the figure.
6) Include the specific values of the controller parameters used in the study.
7) Ensure that Table 1 is presented on a single page and not split between two pages.
8) Address the limitations of the proposed approach in the paper.
N/A.
Author Response
I sincerely appreciate your comments, which have been very beneficial to me, and I hope my reply can meet your requirements.
Reviewer: 3
- The literature can be further improved by refining the research gap and emphasizing the motivation. Additionally, consider providing a comparison table that outlines and compares these techniques.
Reply: According to the suggestions, the introduction section of the literature was rewritten, including the addition of recent literature discussions and contributions of this article, referring to the page 2,3,and a comparison table was added to compare the proposed method with recent works in the same field, referring to Table 1 in page 8.
- Create a flowchart to summarize the proposed approach.
Reply:A flowchart t was created to summarize the design procedure, referring to Fig.11 in page 18.
- Although the paper already presents experimental time-domain results, it might be insightful to include simulation results that explore various scenarios, such as different SCR and Xg/Rg ratios.
Reply:As suggested, the case with different SCR was included into the simulation results,and some discussions were given. The revised content is highlighted in blue in the newly provided manuscript in page 19.
- Provide further elaboration on the mechanism employed (whether it's automatic or manual) for the practical real-time implementation depicted in Figure 15. Additionally, detail the process by which the transition from the value of H from 0.12 to 0.04 is accomplished.
Reply: The further elaboration on the variation of the proportional feedback coefficient H was added into the paper. Additionally, Set H as a global variable, manually change the value of H, and transmit it to the controller through SCI communication.
- For Figure 4, it would be beneficial to use a logarithmic scale. Moreover, for effective illustration, please include both the magnitude and phase in the figure.
Reply: a logarithmic scale is used in Figure 4.Also, due to that some parameters are uncertain, the magnitude is hard to be given. Referring to Fig.4 in page 6.
- Include the specific values of the controller parameters used in the study.
Reply: specific values of the controller parameters was added into the experiment section. The revised content is highlighted in blue in the newly provided manuscript in page 20.
- Ensure that Table 1 is presented on a single page and not split between two pages.
Reply:The format of the Table were adjusted
8) Address the limitations of the proposed approach in the paper.
Reply:The proposed method emulates the ideal differentiator and increases EDR, which is still relatively small. Future research will focus on further improving EDR. The conclusion has been rewritten at the end of the article. The revised content is highlighted in blue in the newly provided manuscript in page 22.
Round 2
Reviewer 1 Report
accept
Author Response
I appreciate your work! thank you very much!
Reviewer 2 Report
There is still some revisions to be made as below;
1. More references must be given in introduction part comparatively
2. How do you derive Equation 1? It must be explained
3. All the figures must be enlarged to be read easily in %100.
4. All the figures must be located correct areas in the text.
5. All the parameters in equation can be given in nomenclature part.
6. Figure 9 must be enlarged and commend more detailed.
7. There are only experimental results. They can be compared with simulations. Also a comparison table for the results. Simulation model must be given.
8. The references must be increased. It is very less for a comprehensive study.
9. The single line diagram of Figure 14 must be given near it to be understand easily.
10. THD analyses can be given more detailed. Also the inrush conditions must be investigated.
Author Response
I sincerely appreciate your comments, which have been very beneficial to me. Due to time constraints, the content was not completely revised based on your comments the last time. This time, I did my best to follow your advice. However, I am unable to answer the No.10 inquirie owing to my restricted ability. I am sorry!!
- More references must be given in introduction part comparatively
Reply: more references were added into introduction part comparatively,it is 40 references in total. The introduction part was revised partly. The revised content is highlighted in blue in the newly provided manuscript in page 2.
- How do you derive Equation 1? It must be explained
Reply: the Equation 1 is a common expression of proportional plus resonance (PR) controller, cited from reference"Stationary frame current regulation of PWM inverters with zero steady state error," the content is as follows:

- All the figures must be enlarged to be read easily in %100.
Reply: All the figures were enlarged to be read in %100.
- All the figures must be located correct areas in the text.
Reply: All the figures were placed after the text.
- All the parameters in equation can be given in nomenclature part.
Reply: a list of the parameters nomenclature was added at the end of the paper.
- Figure 9 must be enlarged and commend more detailed.
Reply: Three images were enlarged and more details were given. Referring to Fig.9 in page 14.
- There are only experimental results. They can be compared with simulations. Also a comparison table for the results. Simulation model must be given.
Reply: simulation results were added in simulation section, the various scenarios, such as different SCR , Xg/Rg ratios and different Lg,etc. The revised content is highlighted in blue in the newly provided manuscript in verification section.
- The references must be increased. It is very less for a comprehensive study.
Reply: the number of the references was increased up to 40 in total.
- The single line diagram of Figure 14 must be given near it to be understand easily.
Reply: the figure 14 was revised. Referring to the figure.14 in submitted manuscript newly.
- THD analyses can be given more detailed. Also the inrush conditions must be investigated.
Reply: The spectra of grid-connected current (THD analysis) were added.referring to the Figure.15d. I am so sorry, I don’t know how to investigate the inrush conditions, but I will study it in the future research as suggested.
Reviewer 3 Report
Thank you to the authors for addressing the majority of my previous comments. There is still one comment that needs to be fully addressed and supported via simulation results: "Although the paper already presents experimental time-domain results, it might be insightful to include simulation results that explore various scenarios, such as different SCR and Xg/Rg ratios."
N/A
Author Response
I sincerely appreciate your comments and your work.
Thank you to the authors for addressing the majority of my previous comments. There is still one comment that needs to be fully addressed and supported via simulation results: "Although the paper already presents experimental time-domain results, it might be insightful to include simulation results that explore various scenarios, such as different SCR and Xg/Rg ratios."
Reply: simulation results were added in simulation section, the various scenarios, such as different SCR , Xg/Rg ratios and different Lg,etc. The revised content is highlighted in blue in the newly provided manuscript in verification section.